# **Evaluation of MUSICA IASI tropospheric water vapour profiles by theoretical error assessments and comparisons to GRUAN Vaisala RS92 measurements**

Christian Borger<sup>1,a</sup>, Matthias Schneider<sup>1</sup>, Benjamin Ertl<sup>1,2</sup>, Frank Hase<sup>1</sup>, Omaira E. García<sup>3</sup>, Michael Sommer<sup>4</sup>, Michael Höpfner<sup>1</sup>, Stephen A. Tjemkes<sup>5</sup>, and Xavier Calbet<sup>6</sup>
<sup>1</sup>Institute of Meteorology and Climate Research (IMK-ASF), Karlsruhe Institute of Technology, Karlsruhe, Germany
<sup>2</sup>Steinbuch Centre for Computing (SCC), Karlsruhe Institute of Technology, Karlsruhe, Germany
<sup>3</sup>Izaña Atmospheric Research Center, Agencia Estatal de Meteorología (AEMET), Santa Cruz de Tenerife, Spain
<sup>4</sup>Deutscher Wetterdienst, Meteorologisches Observatorium Lindenberg, Richard-Aßmann-Observatorium, Am Observatorium 12, 15848, Lindenberg/Tauche, Germany
<sup>5</sup>EUMETSAT, Eumetsat Allee 1, 64295 Darmstadt, Germany
<sup>6</sup>AEMET, C/Leonardo Prieto Castro 8, Ciudad Universitaria, 28071 Madrid, Spain
<sup>a</sup>now at: Satellite Remote Sensing Group, Max Planck Institute for Chemistry, Mainz, Germany *Correspondence to:* C. Borger and M. Schneider

(Christian.Borger@mpic.de and matthias.schneider@kit.edu)

Abstract. Volume mixing ratio water vapour profiles have been retrieved from IASI (Infrared Atmospheric Sounding Interferometer) spectra by using the MUSICA (MUlti-platform remote Sensing of Isotopologues for investigating the Cycle of Atmospheric water) processor. The retrievals are made for IASI observations that coincide with Vaisala RS92 radiosonde measurements performed in the framework of the GCOS (Global Climate Observing System) Reference Upper-Air Network

(GRUAN) in three different climate zones: the tropics (Manus Island, 2°S), mid-latitudes (Lindenberg, 52°N) and polar regions (Sodankylä, 67°N).

The retrievals show good sensitivity with respect to the vertical  $H_2O$  distribution between 1 km above ground and the upper troposphere. Typical DOFS (degrees of freedom for signal) values are about 5.6 for the tropics, 5.1 for summertime midlatitudes, 3.8 for wintertime mid-latitudes, and 4.4 for summertime polar regions. The errors of the MUSICA IASI water

vapour profiles have been theoretically estimated considering the contribution of many different uncertainty sources. For all three climate regions unrecognized cirrus clouds and uncertainties in atmospheric temperature have been identified as the most important error sources and they can reach about 25%.

The MUSICA IASI water vapour profiles have been compared to 100 individual coincident GRUAN water vapour profiles. The systematic difference between the data is within 11% below 12 km altitude; however, at higher altitudes the MUSICA

IASI data show a dry bias with respect to the GRUAN data of up to 21%. The scatter is largest close to the surface (30%), but never exceeds 21% above 1 km altitude. The comparison study documents that the MUSICA IASI retrieval processor provides H<sub>2</sub>O profiles that capture well the large variations in H<sub>2</sub>O volume mixing ratio profiles from 1 km above ground up to altitudes close to the tropopause. Above 5 km the observed scatter with respect to GRUAN data is in reasonable agreement with the combined MUSICA IASI and GRUAN random errors. The increased scatter at lower altitudes might be explained by surface emissivity uncertainties at the summertime continental sites of Lindenberg and Sodankylä and the upper tropospheric dry bias might suggest deficits in correctly modeling the spectroscopic line shapes of water vapour.

### 1 Introduction

30

Atmospheric water plays a key role for the atmospheric energy balance and temperature distribution via radiative effects
(clouds and vapour) and latent heat transport. Hence the distribution and transport of atmospheric moisture is closely linked to atmospheric dynamics on all scales and understanding its spatial and temporal variations is essential for weather and climate modeling. Also, understanding the coupling between moisture transport, clouds and atmospheric dynamics is seen as a major challenge for improving atmospheric models (Stevens and Bony, 2013). In this context the global monitoring of the water vapour distribution is important, whereby the large inhomogeneity in time and space (horizontally and vertically) is particularly
challenging.

In the meantime, several in situ and remote sensing measurement techniques for the observation of water vapour have been established using platforms such as surface stations, balloons, aircraft and satellites. The radiative properties of water vapour enable satellite remote sensing measurements in a large range of wavelength regimes from the visible, e.g. GOME (Grossi et al., 2015), near-infrared, e.g. MODIS (Gao and Kaufman, 2003), thermal infrared, e.g. AIRS (Susskind et al., 2003), TES (Worden

- et al., 2012) and IASI (Herbin et al., 2009; Schneider and Hase, 2011), to the microwave, e.g. AMSU (Rosenkranz, 2000). The instrument IASI (Infrared Atmospheric Sounding Interferometer Clerbaux et al., 2009) aboard EUMETSAT's MetOp satellites is particularly promising: it provides global observations with high resolution and accuracy twice a day on a long-term mission for more than 14 years. Furthermore, IASI follow-up missions have already been approved guaranteeing observations until the 2030s, which will offer great opportunities for studying the atmospheric composition over long time periods.
- When using satellite data in research, it is important to understand their characteristics (sensitivity/representativeness and errors). Theoretical error assessments can be used to reveal the leading error sources. Ideally these error assessments should be accompanied by empirical data validation studies, in which the remote sensing data are compared to independent high quality reference data. Radiosonde measurements are a good candidate for providing references for validating the remote sensing profiles; however, great care is needed for constraining the uncertainties in the radiosonde data (McMillin et al., 2007).
- In particular promising in this context are the temperature and humidity profiles produced from Vaisala RS92 radiosonde measurements in the framework of the GCOS Reference Upper-Air Network (GRUAN, www.gruan.org), a subnetwork of the Global Climate Observing System (GCOS, ://www.wmo.int/pages/prog/gcos/index.php). Currently GRUAN consists of about 30 reference sites and provides humidity and temperature profiles of a high and well documented quality (Dirksen et al., 2014).

In this paper we perform a detailed theoretical error assessment and an empirical validation of the water vapour profiles as generated by the MUSICA (MUlti-platform remote Sensing of Isotopologues for investigating the Cycle of Atmospheric

water Schneider et al., 2016) IASI retrieval processor. The retrievals are made for three different climate regions (tropics, midlatitudes, polar regions) and for coincidences with GRUAN in situ radiosonde measurements, which we use as the reference for the empirical validation study. Our investigations will give an overview on the retrieval's capability of profiling atmospheric water vapour. The paper is organized as follows: Section 2 will give a brief overview of the MUSICA IASI processor by describing general retrieval and error estimation principles, by presenting the particularities of the MUSICA retrieval setup and by discussing the MUSICA retrieval output. Section 3 presents the sites and time periods for which the data evaluation is performed. Section 4 shows the theoretical IASI data characterisation and Sect. 5 presents and discusses the results of the

5 comparison between the remote sensing data and the GRUAN in situ reference data. In Sect. 6 we summarize the outcomes of the study.

### 2 MUSICA IASI data

### 2.1 Atmospheric remote sensing retrieval principles

In this subsection we give a very brief introduction to the principles of the optimal estimation retrieval method. It is a standard retrieval method in atmospheric remote sensing. For more details please refer to Rodgers (2000) and for a general introduction on vector and matrix algebra dedicated textbooks are recommended.

Atmospheric remote sensing means that the atmospheric state is retrieved from the radiation measured after having interacted with the atmosphere. This interaction of radiation with the atmosphere is modeled by a radiative transfer model (also called the forward model, F), which enables relating the measurement vector and the atmospheric state vector by:

$$15 \quad \boldsymbol{y} = \boldsymbol{F}(\boldsymbol{x}, \boldsymbol{b}). \tag{1}$$

We measure y (the measurement vector, e.g. a thermal nadir spectrum in the case of IASI) and are interested in x (the atmospheric state vector). Vector b represents auxiliary parameters (like surface emissivity) or instrumental characteristics (like the instrumental line shape), which are not part of the retrieval state vector. However, a direct inversion of Eq. (1) is generally not possible, because there are many atmospheric states x that can explain one and the same measurement y.

For solving this ill-posed problem a cost function J is set up, that combines the information provided by the measurement with a priori known characteristics of the atmospheric state:

$$J = [\boldsymbol{y} - \boldsymbol{F}(\boldsymbol{x}, \boldsymbol{b})]^T \mathbf{S}_{\mathbf{y}, \text{noise}}^{-1} [\boldsymbol{y} - \boldsymbol{F}(\boldsymbol{x}, \boldsymbol{b})] + [\boldsymbol{x} - \boldsymbol{x}_a]^T \mathbf{S}_a^{-1} [\boldsymbol{x} - \boldsymbol{x}_a].$$
(2)

Here, the first term is a measure of the difference between the measured spectrum (represented by y) and the spectrum simulated for a given atmospheric state (represented by x), while taking into account the actual measurement noise (S<sub>y,noise</sub> is the measurement noise covariance matrix). The second term of the cost function (Eq. 2) constrains the atmospheric solution state (x) towards an a priori most likely state (x<sub>a</sub>), whereby the kind and strength of the constraint are defined by the a priori covariance matrix S<sub>a</sub>. The constrained solution is reached at the minimum of the cost function (Eq. 2). Due to the nonlinear behavior of F(x, b), the minimisation is generally achieved iteratively. For the (i + 1)th iteration it is:

$$\boldsymbol{x_{i+1}} = \boldsymbol{x_a} + \mathbf{G}_{\mathbf{i}}[\boldsymbol{y} - \boldsymbol{F}(\boldsymbol{x_i}, \boldsymbol{b}) + \mathbf{K}_{\mathbf{i}}(\boldsymbol{x_i} - \boldsymbol{x_a})]. \tag{3}$$

30 K is the Jacobian matrix (derivatives that capture how the measurement vector will change for changes in the atmospheric state *x*). G is the gain matrix (derivatives that capture how the retrieved state vector will change for changes in the measurement

vector y). G can be calculated from K,  $S_{y,noise}$  and  $S_a$  as:

$$\mathbf{G} = (\mathbf{K}^T \mathbf{S}_{\mathbf{y}, \mathbf{noise}}^{-1} \mathbf{K} + \mathbf{S}_{\mathbf{a}}^{-1})^{-1} \mathbf{K}^T \mathbf{S}_{\mathbf{y}, \mathbf{noise}}^{-1}.$$
(4)

The averaging kernel is an important component of a remote sensing retrieval and it is calculated as:

$$5 \quad \mathbf{A} = \mathbf{G}\mathbf{K}.$$

The averaging kernel A reveals how a small change of the real atmospheric state vector x affects the retrieved atmospheric state vector  $\hat{x}$ :

$$\hat{\boldsymbol{x}} - \boldsymbol{x}_a = \mathbf{A}(\boldsymbol{x} - \boldsymbol{x}_a). \tag{6}$$

The propagation of errors due to parameter uncertainties  $\Delta b$  can be estimated analytically with the help of the parameter 10 Jacobian matrix  $\mathbf{K}_{\mathbf{b}}$  (derivatives that capture how the measurement vector will change for changes in the parameter  $\boldsymbol{b}$ ). According to Eq. (3), using the parameter  $\boldsymbol{b} + \Delta b$  (instead of the correct parameter  $\boldsymbol{b}$ ) for the forward model calculations will result in an error in the atmospheric state vector of:

$$\Delta \hat{x} = -\mathbf{G} \mathbf{K}_{\mathbf{b}} \Delta b. \tag{7}$$

The respective error covariance matrix  $S_{\hat{x},b}$  is:

$$\mathbf{S}_{\hat{\mathbf{x}},\mathbf{b}} = \mathbf{G}\mathbf{K}_{\mathbf{b}}\mathbf{S}_{\mathbf{b}}\mathbf{K}_{\mathbf{b}}^{T}\mathbf{G}^{T},$$
 (8)

where  $\mathbf{S}_{\mathbf{b}}$  is the covariance matrix of the uncertainties  $\Delta b$ .

Noise on the measured radiances also affects the retrievals. The error covariance matrix for noise can be analytically calculated as:

$$\mathbf{S}_{\hat{\mathbf{x}},\mathbf{noise}} = \mathbf{G}\mathbf{S}_{\mathbf{y},\mathbf{noise}}\mathbf{G}^T,\tag{9}$$

where  $S_{y,noise}$  is the covariance matrix for noise on the measured radiances y.

## 2.2 The MUSICA retrieval setup

The MUSICA IASI retrieval is based on a nadir version of the retrieval code PROFFIT (PROFile FIT Hase et al., 2004) and on the corresponding radiative transfer model PRFFWD (PRoFit ForWarD model Hase et al., 2004; Schneider and Hase, 2009). The nadir code has been developed in support of the project MUSICA (MUlti-platform remote Sensing of Isotopologues for

investigating the Cycle of Atmospheric water, http://www.imk-asf.kit.edu/english/musica.php). The PRFFWD-nadir code has been recently updated by including water continuum calculations according to the model "MT\_CKD" v2.5.2 (Delamere et al., 2010; Payne et al., 2011; Mlawer et al., 2012).

For the MUSICA IASI retrieval calculations a single broad spectral window from  $1190 \text{ cm}^{-1}$  to  $1400 \text{ cm}^{-1}$  is used. The spectral signatures of  $H_2^{16}O$ ,  $H_2^{18}O$  and  $H_2^{17}O$  are fitted together as a single species (from now on called  $H_2O$ ) and  $^{1}H^{2}H^{16}O$ 

(from now on called HDO) as a separate species. Furthermore, the retrieval's spectral window contains spectroscopic features of  $CH_4$  and  $N_2O$  as well as weak spectroscopic features of  $HNO_3$  and very weak spectroscopic features of  $CO_2$ . All these trace gases are simultaneously fitted during the retrieval process whereby the spectroscopic parameters are taken from the HITRAN

2016 database (Gordon et al., 2017) with small modifications for HDO parameters (similar to Schneider et al., 2016, the line intensity parameters of HDO have been increased by 10%).

For the water isotopologues,  $CH_4$ ,  $N_2O$  and  $HNO_3$  profile retrievals are performed on a logarithmic scale. For  $CO_2$  the a priori profiles are scaled. A single a priori profile is used for all the retrievals for each of the different trace gases, i.e. the used a priori are the same for all locations and time periods (Schneider et al., 2016; García et al., 2017). For  $CH_4$ ,  $N_2O$ ,  $HNO_3$  and

10 CO<sub>2</sub> the a priori profiles are averaged low latitude profiles from WACCM (Whole Atmosphere Community Climate Modelversion 5, http://waccm.acd.ucar.edu) and are provided by NCAR (National Center for Atmospheric Research, J. Hannigan, private communication). The water vapour isotopologue a priori data are averages obtained from the isotopologue incorporated global general circulation model LMDZ (Risi et al., 2012).

The retrieval also fits the surface temperature and the atmospheric temperature profile, whereby the a priori temperatures are

15 taken from the EUMETSAT IASI level 2 (L2) products. There is no constraint on the surface temperature. The allowed atmospheric temperature variations are 1 K at ground, 0.5 K in the free troposphere, and 0.75 K above the tropopause. This altitude dependency follows roughly the altitude dependency of uncertainties in the EUMETSAT IASI L2 atmospheric temperature profiles (August et al., 2012).

The MUSICA IASI water vapour retrieval only works for pixels that are not contaminated by clouds, whereby we rely on

- the IASI L2 cloud flag (we require zero for the flag "cldfrm"). Ground elevations are from GTOPO30 developed by the US Geological Survey and provided by the Oak Ridge National Laboratory Distributed Active Archive Center (ORNL DAAC). GTOPO30 is a global digital elevation model with a horizontal grid spacing of 30-arc seconds (approximately 1 km). The land surface emissivities are from the "global database of infrared land surface emissivity" (IREMIS; http://cimss.ssec.wisc.edu/ iremis/; Seemann et al., 2008) and the sea surface emissivities are calculated according to the model of Masuda et al. (1988).
- Figure 1 depicts an example of a typical radiance spectrum in the retrieval's spectral range as measured by IASI (upper graph) and the corresponding differences compared to the simulated spectra (the residuals, lower graph). The residuals are mostly within the order of the instrument's  $1\sigma$  measurement noise (Pequignot et al., 2008). However, there are also distinctive spectral signatures that are not well understood, specifically at 1250 cm<sup>-1</sup> and at 1280 cm<sup>-1</sup>.

For further information on the retrieval setup and its evolution more detailed descriptions are available in Schneider and 30 Hase (2011), Wiegele et al. (2014), Schneider et al. (2016) and (García et al., 2017).

### 2.3 The MUSICA retrieval output

The output of the retrieval refers to the  $\{\ln[H_2O], \ln[HDO]\}$  basis system. In this basis system the state vector x consists of the vector for the H<sub>2</sub>O profile extended by the vector for the HDO profile:

$$\boldsymbol{x} = \begin{pmatrix} \boldsymbol{x}_{\mathrm{H}_{2}\mathrm{O}} \\ \boldsymbol{x}_{\mathrm{HDO}} \end{pmatrix}.$$
 (10)

Correspondingly, the averaging kernel matrix A has  $2 \times 2$  blocks

5 
$$\mathbf{A} = \begin{pmatrix} \mathbf{A}_{11} & \mathbf{A}_{12} \\ \mathbf{A}_{21} & \mathbf{A}_{22} \end{pmatrix}.$$
 (11)

 $A_{11}$  and  $A_{22}$  describe how the retrieved H<sub>2</sub>O and HDO states depend on the actual atmospheric H<sub>2</sub>O and HDO variations, respectively, and  $A_{12}$  and  $A_{21}$  reveal the cross-dependencies of the retrieved H<sub>2</sub>O on the actual atmospheric HDO and of the retrieved HDO on the actual atmospheric H<sub>2</sub>O, respectively. Since H<sub>2</sub>O and HDO vary largely in parallel, we use in the following the  $A_{11} + A_{12}$  as the kernel for H<sub>2</sub>O (see also Sect. 4.3 in Barthlott et al., 2017).

Similarly, retrieval error covariance matrices consist of  $2 \times 2$  blocks where the blocks in the diagonal represent the H<sub>2</sub>O and HDO covariances. For this study only the H<sub>2</sub>O covariance block is of interest (i.e. we are only interested in the H<sub>2</sub>O error covariances). The outer diagonal blocks represent the error covariances between H<sub>2</sub>O and HDO.

### 3 Reference data and sites

The theoretical and empirical assessment studies are made for cloud-free IASI measurements that coincide with GRUAN processed Vaisala RS92 radiosonde measurements. Useful coincidences are defined in accordance to Pougatchev et al. (2009) and Calbet et al. (2017).

We identified three different sites with coincidence between IASI and GRUAN measurements: Manus Island (Papua New Guinea; 2°5'S, 146°58'E) for the tropics, Lindenberg (Germany; 52°12'N, 14°7'E) for the mid-latitudes and Sodankylä (Finland; 67°25'N, 26°35'E) for the polar region.

- Figure 2 depicts all the GRUAN H<sub>2</sub>O profiles that coincide with IASI observations made for cloud-free conditions. There are 25 individual GRUAN profiles for Manus Island (during different months in 2011-2013), 58 for Lindenberg (26 during summer 2007 and 32 during different months in 2008), and 17 for Sodankylä (during summer 2007), i.e. in total there are 100 individual GRUAN radiosonde measurements that coincide with IASI cloud-free measurements. These four ensembles of GRUAN profiles are well representative for the highly varying tropospheric H<sub>2</sub>O distributions. In the free middle/upper
- 25 troposphere the data show variations of up to two orders of magnitude. At the tropical site of Manus Island we observe up to 10 000 ppmv (at 5 km a.s.l.) and up to 1000 ppmv (at 10 km a.s.l.), whereas at the mid-latitude and polar sites of Lindenberg and Sodankylä the H<sub>2</sub>O concentrations can be as small as 100 ppmv and 10 ppmv, respectively. In this context using the four ensembles of GRUAN data enables us to make an evaluation of the retrieval performance that has a good global validity.

The coincidences at the three sites are for different time periods and there is not a strictly uniform data set for creating the retrieval input files: EUMETSAT L2 data are not available for all the time periods or are generated by different EUMETSAT L2 product processing facility (PPF) software version (for more details see Sects. 3.2-3.4 and the summary of Table 1).

### 3.1 GRUAN processed Vaisala RS92 in-situ profiles

The Vaisala RS92 radiosonde is equipped with a wire-like capacitive temperature sensor ("Thermocap"), two polymer capacitive moisture sensors ("Humicap"), a silicon-based pressure sensor and a GPS receiver to measure position, altitude and winds.
Each second the RS92 transmits sensor data, which are received, processed and stored by the ground station equipment.
The Humicap consists of a hydro-active polymer thin film as dielectric between two electrodes applied on a glass substrate. The humidity sensors are not covered by protective caps, but they are alternately heated to prevent icing. To prevent overheating, the heating of the humidity sensors is switched off below –60 °C, or above 100 hPa, whichever is reached first. Humicaps show

10 good performance over a wide range of temperatures but suffer from systematic errors such as dry bias due to solar radiative heating and a response lag below -40 °C. Known main error sources affecting the humidity profile are daytime solar heating of the Humicaps introducing a dry bias, sensor time-lag at temperatures below about -40 °C and temperature-dependent calibration correction.

We work with Vaisala RS92 data that have been processed by the GRUAN lead centre (http://www.gruan.org). The GRUAN
data processing assures that the obtained humidity, pressure and temperature profiles are well-calibrated and highly accurate (Dirksen et al., 2014: Sommer et al., 2016).

### 3.2 Manus Island (MI)

20

At Manus Island we have coincidences in 2011, 2012 and 2013 with 25 individual GRUAN radiosonde profiles. The collocation of IASI and GRUAN measurements has been performed by EUMETSAT in the framework of a planned IASI retrieval comparison study (Calbet et al., 2017) allowing a spatial and temporal window of 25 km and 30 min respectively.

For our retrieval we use the a priori temperatures (atmosphere and surface skin) as well as surface emissivities from the EUMETSAT IASI L2 product generated with the IASI L2 PPF software version 5. Since most of the ground scenes are over the ocean surface, the emissivity values are mainly according to the model of Masuda et al. (1988). The satellite pixels have been careful examined for clouds by EUMETSAT according to the cloud flags as provided in the IASI L2 data and in addition by visual inspection (Calbet et al. 2017).

25 by visual inspection (Calbet et al., 2017).

### 3.3 Lindenberg 2008 (LI08)

For Lindenberg there are coincidences with 32 individual GRUAN profiles in 2008 (representative of all seasons). We performed the collocation and required that the satellite pixel has to be within a distance of 25 km with respect to the starting position of the radiosonde and that the satellite's pixel sensing time has to be within the sensing time period of the radiosonde. 30 As for Manus Island we rely on the IASI L2 data for our retrieval input data (surface and atmospheric temperatures, surface emissivity, cloud filter, etc.). However, while for the 2011-13 time period (Manus Island) the IASI L2 data are generated with the IASI L2 PPF software version 5, for the 2008 retrievals we work with L2 data generated by the IASI L2 PPF software version 4. As shown in García et al. (2017) there can be inconsistencies between the MUSCA IASI products that are generated using different IASI L2 PPF software versions.

#### 5 **3.4** Lindenberg 2007 (LI07) and Sodankylä 2007 (SK07)

In 2007 we have 26 individual GRAUN profiles for Lindenberg and 17 individual GRUAN profiles for Sodankylä (details on the Sodankylä campaign are available in Calbet et al., 2011) that coincide with IASI observations. This dataset is limited to the summer observations. We performed the collocation using the same criteria as for the Lindenberg 2008 coincidences, i.e. we required that the satellite pixel has to be within a distance of 25 km with respect to the starting position of the radiosonde and that the satellite's pixel sensing time has to be within the sensing time period of the radiosonde.

For summer 2007 IASI L2 data have not been available for our retrieval input. Thus data from the radiosonde measurements have been used as the a priori temperatures. Above the top altitude of the radiosonde we use zonally and monthly averaged temperature climatologies (COSPAR International Reference Atmosphere Rees et al., 1990). Using the GRUAN temperatures (and climatologies at higher altitudes) instead of the IASI L2 temperatures as the a priori for the atmospheric temperatures

15 might cause some inconsistency between the LI07 and SK07, on the one hand, and the MI and LI08 retrievals, on the other hand.

Surface emissivities are taken from the "global database of infrared land surface emissivity" (IREMIS; http://cimss.ssec. wisc.edu/iremis/; Seemann et al., 2008), i.e. in agreement to the retrievals for 2008 (IASI L2 emissivities are based on IREMIS for land surfaces and use the Masuda model for sea surfaces). Because there are no IASI L2 cloud products for summer 2007,

we use the radiosonde measurements and the cloud detection algorithm according to the model of Zhang et al. (2010) for 20 identifying cloud-free situations.

### **Theoretical MUSICA IASI data characterisation** 4

### **Averaging Kernels** 4.1

10

Figure 3 illustrates examples of  $H_2O$  row kernels ( $A_{11} + A_{12}$ , logarithmic state vector entries according to Sect. 2.3) for the three reference sites. Grey lines show all row kernels and the thick coloured lines highlight the kernels for some selected 25 altitudes. The peaks of the row kernels for the altitudes of 1.8 km, 3.6 km, 6.4 km and 9.8 km are close to their nominal altitudes, meaning that the values retrieved for these altitudes well represent the situation at the nominal altitude. In contrast the row kernel for the ground altitude peaks typically 1 km above ground altitude, meaning that data retrieved at the ground level mainly represent the altitudes about 1 km above the ground level. The row kernel for the 13.6 km altitude peaks close 30

to its nominal altitude only for the tropical site of Manus Island. At Lindenberg and Sodankylä the respective kernels peak at

9-10 km altitude, i.e. at these sites variations in the 13.6 km retrieval data are mainly driven by the actual atmospheric variations at 9-10 km.

The thick black dashed line represents the sum along the row of the averaging kernel matrix and is a measure for the remote sensing systems sensitivity. This value is typically between 0.9 and 1.1 from 1 km up to 11 km at Lindenberg and Sodankylä, and from 1 km up to 13 km at Manus Island.

- The seasonal dependency of the averaging kernels is indicated in Fig. 4, which depicts the seasonal variations in the degree of freedom for signal (DOFS) values. The DOFS values are calculated as the trace of the averaging kernel matrix and the higher the DOFS values the more profile information is in the retrieved atmospheric state. In the tropics we observe no seasonal dependency. In the mid-latitudes the DOFS values are distinctively higher in summer than in winter. More details on this seasonal dependency are provided in Fig. 5, which depicts typical wintertime and typical summertime row kernels
- for Lindenberg. It seems that the seasonal variation is mostly a variation of the sensitivity at higher altitudes. In summer the remote sensing system can detect the actual atmospheric variations up to 11-12 km, whereas in winter the sensitivity is limited to altitudes below 8-9 km. This is connected to the variation of the tropopause altitude. As shown in Schneider et al. (2017) the averaging kerneln depend strongly on the atmospheric temperature and humidity profiles (as well as on the surface temperature and emissivity).
- In summary, at all three different sites the MUSICA IASI retrieval provides H<sub>2</sub>O profile information from about 1 km above ground up to about the tropopause altitude.

### 4.2 Calculation of error Jacobians

The error Jacobians ( $\mathbf{K}_{\mathbf{b}}$  from Eqs. 7 and 8) are calculated by the forward model PRFFWD as follows: PRFFWD is executed running on a vertical grid of 28 levels from surface altitude to approximately 55 km above mean sea level. For every site 20 reference forward calculations are performed for all cloud-free situations. The input (i.e. temperature, trace gas concentrations, etc.) for the reference forward model runs is the same as the input used in the forward calculation of the last iteration step of the MUSICA IASI retrievals, i.e. the reference radiances are given by  $F(\hat{x}, b)$ . Then for each reference scenario we make additional forward calculations with slightly modified parameters, i.e. we calculate  $F(\hat{x}, b + \Delta b)$ . For a measurement vector yhaving m elements and a parameter vector b having n elements the Jacobian matrix  $\mathbf{K}_{\mathbf{b}}$  will have the dimension  $m \times n$ . The individual matrix elements are calculated as:

$$\mathbf{K}_{\mathbf{b}_{\mathbf{k},\mathbf{l}}} = \frac{F_k(\hat{\boldsymbol{x}}, \boldsymbol{b} + \Delta b_l) - F_k(\hat{\boldsymbol{x}}, \boldsymbol{b})}{\Delta b_l},\tag{12}$$

where k is the index for the kth element of the measurement state vector y (simulated by vector function F) and l is the index for the lth element of the parameter vector b, respectively.

Table 2 gives an overview of the uncertainty assumptions ∆b used for calculating the Jacobians and for performing the error
setimation. The calculations of the error Jacobians for water vapour continuum and clouds require a specific treatment, which is detailed in the following two subsections.

### 4.2.1 Water vapour continuum

We assume that calculations based on the model "MT\_CKD" v2.5.2 (Mlawer et al., 2012; Delamere et al., 2010; Payne et al., 2011) only partly capture the full water vapour continuum effect. For the respective Jacobian calculation we perform forward calculations without considering the water vapour continuum ( $F_{noWVC}(\hat{x}, b)$ ). Then we calculate the Jacobian matrix as  $K_{noWVC} = F_{noWVC}(\hat{x}, b) - F(\hat{x}, b)$ . The spectral response for an underestimation of 10% of the water vapour continuum effect is then  $K_{noWVC}\Delta b_{noWVC}$  with  $\Delta b_{noWVC} = 0.1$ .

### 4.2.2 Opaque clouds (cumulus)

5

We estimate the influence of fractional coverage by opaque liquid cumulus clouds with different cloud top altitudes (1.3 km, 3.0 km and 4.9 km). The radiance at top of the cloudy atmosphere  $F_{cum}(\hat{x}, b)$  is calculated by starting PRFFWD at the cloud's top height, assuring that no radiation from below the cloud contributes to  $F_{cum}(\hat{x}, b)$ . Additionally it is assumed that the surface

10 emissivity of the cloud is 1.0 and that the skin temperature of the cloud's upward looking surface is in thermal equilibrium with the surrounding air temperature. The Jacobian matrix for opaque cumulus clouds is then  $\mathbf{K}_{cum} = \mathbf{F}_{cum}(\hat{\mathbf{x}}, \mathbf{b}) - \mathbf{F}(\hat{\mathbf{x}}, \mathbf{b})$  and the spectral response of a 10% fractional cloud cover is  $\mathbf{K}_{cum} \Delta b_{cum}$  with  $\Delta b_{cum} = 0.1$ .

### 4.2.3 Transmitting clouds (mineral dust and cirrus)

Some clouds are not opaque and we have to consider partial attenuation by the cloud particles. This is the case for cirrus clouds and mineral dust clouds. We consider these clouds by introducing them as an additional species in the forward model calculations. The extinction of these clouds is the sum of absorption and scattering. Since PRFFWD does not include the simulation of scattering clouds we calculate the attenuated radiances using forward model calculations from KOPRA (Karlsruhe optimized and precise radiative transfer algorithm; Stiller, 2000) and consider single scattering.

The frequency dependency of the extinction cross sections, the single scattering albedo, and the scattering phase functions of the clouds are calculated from OPAC v4.0b (Optical Properties of Aerosol and Clouds; Hess et al., 1998; Koepke et al., 2015). For cirrus clouds we assume the particle composition as given by OPAC's "Cirrus 3" ice cloud example (see Table 1b in Hess et al., 1998) and for mineral dust clouds a particle composition according to OPAC's "Desert" aerosol composition example (see table 1able 4 in Hess et al., 1998).

We make cirrus cloud forward calculations  $F_{cir}$  considering cirrus clouds with a vertical cloud layer thickness of 1 km and 25 cloud top at different altitudes ranging from 6 km to 14 km. The Jacobians are calculated as  $\mathbf{K}_{cir} = F_{cir}(\hat{x}, b) - F(\hat{x}, b)$  and for a cloud coverage of 50% the spectral response is  $\mathbf{K}_{cir} \Delta b_{cir}$  with  $\Delta b_{cir} = 0.5$ .

For the dust clouds we make forward calculations  $F_{dust}$  for homogeneous 2 km thick layers between the ground and 6 km altitude. The Jacobians are then given as  $\mathbf{K}_{dust} = F_{dust}(\hat{x}, b) - F(\hat{x}, b)$ .

### 4.3 Spectral response on uncertainty

Figure 6 depicts the spectral responses (i.e.  $K_b\Delta b$ ) for an example of different uncertainty sources for a typical situation at the tropical reference site. The left panel shows that lower tropospheric temperature uncertainties mainly affect the spectra between  $1190 \text{ cm}^{-1}$  to  $1250 \text{ cm}^{-1}$  (which is also the spectral region of an "atmospheric window"), but is negligible for higher wavenumbers. This is in contrast to upper tropospheric temperature uncertainties, which have highest spectral responses for wavenumbers larger than  $1250 \text{ cm}^{-1}$ .

The right panel of Fig. 6 illustrates that uncertainties in dust layers and uncertainties due to cirrus clouds have the highest impact at the lower end of wavenumbers and that a cirrus cloud has a different dependency on wavenumber than a dust layer. Furthermore unrecognized clouds have the opposite effect on the spectrum than increasing the atmospheric temperatures although affecting the spectrum in the same order of magnitude.

### 10 4.4 Estimated errors

5

15

Figure 4 shows a certain seasonal variability in the DOFS values (in particular at the mid-latitude site), indicating varying sensitivities of the remote sensing system. This variation is also present in the sensitivity with respect to uncertainty sources. For this reason we present the estimated errors for all the Manus Island and Sodankylä retrievals (MI and SK07) and for all the Lindenberg 2008 retrievals (LI08). The Lindenberg 2008 error estimations are representative for all seasons, hence they well cover the full sensitivity variation with respect to uncertainty sources.

Table 2 gives on overview on the the different uncertainty sources we consider for our error estimation. We distinguish between random uncertainty sources (the uncertainty affecting an observation is uncorrelated with the uncertainty affecting another observation), systematic uncertainty sources (the uncertainty is the same for all observations) and uncertainty sources that are always positive but with a random amplitude (clouds: the sky is either cloud free or covered by a random amount of clouds).

## 20 clouds).

### 4.4.1 Errors caused by random uncertainty sources

Figure 7 depicts from the top to the bottom the  $H_2O$  error profiles due to the random uncertainties instrumental noise, emissivity and atmospheric temperatures (from the left to the right for Manus Island, Lindenberg, and Sodankylä). The error profiles shown are the square root of the diagonal elements of the error covariance matrix  $S_{\hat{x},noise}$  calculated for the instrumental noise

according to Eq. (9) and of the error covariance matrix  $S_{\hat{x},b}$  calculated for emissivity and atmospheric temperature according to Eq. (8).

For the calculations of  $S_{\hat{x},noise}$  we assume a noise covariance  $S_{y,noise}$  of the IASI radiances according to Pequignot et al. (2008). The measurement noise errors vary around 2-10% near the ground, but decrease to approximately 2-3% above the boundary layer and remain there throughout the free troposphere. Close to the tropopause errors increase again to values of

30 around 10%. For Manus Island we observe similar errors for all the different observations. For Sodankylä and in particular for Lindenberg the errors vary. For instance, in the lower troposphere at Lindenberg the error is 10% for some days, but only 1-3% for other days. The varying sensitivity with respect to the uncertainty sources is due to the varying atmospheric conditions and in agreement with the varying DOFS values as documented by Fig. 4 (the Lindenberg data cover all mid-latitude seasons).

For the calculating the error covariances  $S_{\hat{x},b}$  due to surface emissivity uncertainties, we assume a 1% emissivity uncertainty and a spectral correlation length of this uncertainty of  $100 \text{ cm}^{-1}$ . The resulting errors are highest close to the ground and for the continental sites of Lindenberg and Sodankylä, where they can reach 30%. Above 5 km altitude these errors are generally below 2%.

For the calculating the error covariances  $S_{\hat{x},b}$  due to atmospheric temperatures uncertainties, we assume 2 K uncertainty from ground-2 km and 1 K uncertainty above 2 km altitude with correlation lengths increasing from 2 km at ground to 10 km in the stratosphere. The errors are typically 10-15%, but can occasionally reach 25%.

### 4.4.2 Errors caused by systematic uncertainty sources

5

25

10 Fig. 8 shows from the top to the bottom the H<sub>2</sub>O error profiles due to systematic uncertainties in surface emissivity, atmospheric temperature and spectroscopic parameters. The error profiles are calculated as  $\Delta \hat{x}$  according to Eq. (7).

We assume to patterns of surface emissivity uncertainty. The first pattern means a -1% uncertainty at the spectral grid points  $1185 \text{ cm}^{-1}$  and  $1240 \text{ cm}^{-1}$  and 0% uncertainty at the grid points  $1295 \text{ cm}^{-1}$ ,  $1350 \text{ cm}^{-1}$  and  $1405 \text{ cm}^{-1}$  (this means that between  $1240 \text{ cm}^{-1}$  and  $1295 \text{ cm}^{-1}$  the uncertainty is linearly changing from -1% to 0%). The second pattern means a

- 15 -1% uncertainty at the spectral grid points  $1405 \text{ cm}^{-1}$  and  $1350 \text{ cm}^{-1}$  and 0% for the rest (with a linear change -1% to 0% between  $1350 \text{ cm}^{-1}$  and  $1295 \text{ cm}^{-1}$ ). The top row of panels in Fig. 8 shows that surface emissivity uncertainties are mainly important for the wavenumber region below  $1300 \text{ cm}^{-1}$  (the first uncertainty pattern). An emissivity uncertainty of -1% has a rather uniform response at Manus Island: positive error of up to 5% close to the ground and a weak negative error around 3 km altitude. At the continental sites of Lindenberg and Sodankylä the response on a systematic -1% emissivity uncertainty can be
- 20 positive or negative close to the ground (it varies between about -25% and +20%). Around 3 km the error response is generally negative and between -2% and -20%.

Positive atmospheric temperature uncertainties cause large positive errors in the retrieved tropospheric  $H_2O$  profiles (we assume a systematic uncertainty of +2 K up to 2 km altitude and +1 K at higher altitudes). The errors can reach +30%, whereby these errors are largest for the atmospheric layers where the atmospheric temperature uncertainty is assumed. For instance, uncertainties in lower tropospheric temperature (ground-2 km, black lines) cause maximal errors from ground up to 3 km and decrease rapidly with altitude onwards, whereas uncertainties in upper tropospheric temperature (5-10 km, green lines) are negligible from ground up to 6 km, but then increase to values of around +20% at 8 km.

Concerning spectroscopic parameters we consider systematic uncertainties in the  $H_2O$  line intensity and pressure broadening parameters and an uncertainty in the applied water continuum model. The uncertainty in the water vapour continuum model

30 causes error profiles having small oscillations. For a water continuum model that underestimates the water continuum effect by 10% (see Sect. 4.2.1), the error is positive near ground (about +2%), negative at around 3 km altitude (about -4%) and negligible for altitudes above 5 km. A positive uncertainty of +5% in the water vapour (H<sub>2</sub>O and HDO) line strength parameter causes a negative error of about -5% in the retrieved H<sub>2</sub>O values. The impact of +5% uncertainties in the pressure broadening parameter depends on the reference site: at Manus Island the resulting errors are negligible above 3 km, but at Lindenberg and Sodankylä the error profiles contain strong oscillations with maximal error of about +10% above 10 km altitude.

This behavior of the errors due to uncertainties in the line shape modeling might be explained as follows: most of thermal nadir spectra's information about the vertical  $H_2O$  distribution is a consequence of the vertical atmospheric gradients of temperature and humidity. Without this gradients the spectral emission from a lower atmospheric layer is widely canceled out by

- the absorption at a higher layer. The gradients are generally strong up to the tropopause, i.e. up to the tropopause the remote sensing system's sensitivity is widely determined by these gradients. At Manus Island the tropopause is generally above 15 km, whereas at Lindenberg and Sodankylä is can be at much lower altitudes. This can be observed in Fig.2, which indicates a decrease of H<sub>2</sub>O concentration up to 16 km above Manus Island, but only up to about 13 km and 12 km above Lindenberg and Sodankylä, respectively. Due to the weaker gradients above Lindenberg and Sodankylä and the relatively good spectral
- resolution of the IASI spectra, the line shapes do also provide information on the vertical distribution of  $H_2O$ . This is due to the pressure broadening effect, i.e. the broadness of the line decreases with decreasing pressure. As a consequence the  $H_2O$ profiles retrieved at Lindenberg and Sodankylä are much more affected by uncertainties in the line shape modeling than the profiles retrieved at Manus Island.

### 4.4.3 Errors due to unrecognized clouds

- Figure 9 shows the influence of different cloud types on the errors. Uncertainties due to unrecognized cirrus clouds (top row in Fig. 9) lead to errors of -20% from 3-6 km at all sites and then decrease with altitude. However their impact on the WVMR profiles in the boundary layer shows large variation, especially at Lindenberg and Sodankylä, which is a result of the more variable atmospheric conditions at these sites (compared to the tropical site of Manus Island).
- The influence of a 10% fractional cloud cover of opaque clouds depends on the height where the clouds are assumed (middle row in Fig. 9): Clouds at 1.3 km show only a small impact on the humidity profiles in the boundary layer with error magnitudes of 5-10%, but clouds at 3.0 km account for errors of more than 10% up to 5 km above mean sea level. Yet similarly to cirrus clouds their effect in the boundary layer shows large variation at Lindenberg and Sodankylä.

The error profiles due to mineral dust layers (bottom row in Fig. 9) show that such layers have almost no impact if they are situated in the boundary layer, however if they are situated in the middle troposphere the errors are more than 10%. The effect of dust clouds can be in particular large for the mid-latitude site of Lindenberg, where we also observe the largest variability in

of dust clouds can be in particular large for the mid-latitude site of Lindenberg, where we also observe the largest variability in the calculated error profiles.

### 5 Comparison of GRUAN and IASI data

We use GRUAN processed Vaisala RS92 radiosonde measurements as reference for empirically validating the retrieved MU-SICA IASI  $H_2O$  profiles. The radiosonde ascents are collocated temporally and spatially with MetOp overpasses (for details see Sect. 3), which is essential for a meaningful comparison.

### 5.1 Regridding and smoothing of the high resolution GRUAN in-situ profiles

The in situ profiles have a high vertical resolution. This differs from the remote sensing profiles, which can only detect the major characteristics of the vertical  $H_2O$  distribution. Before comparing the data we have to account for these different characteristics by regridding and smoothing the in situ profiles.

While the remote sensing retrieval provides atmospheric states and averaging kernels on a coarse atmospheric grid (between ground level and about 55 km a.s.l. 28 grid points are defined), the radiosonde reports data about every 5 m. So we have to regrid the radiosonde data to the coarse vertical grid used by the remote sensing retrieval. In order to guarantee that the regridding does not significantly affect the H<sub>2</sub>O partial columns, the regridding is performed in two steps.

First, the radiosonde data points between the 28 MUSICA retrieval grid points are averaged by using a triangle inversedistance weighted function resulting in a first estimate of the regridded radiosonde data. In the second step this first estimate

- 10 is corrected by requiring that the partial columns between adjacent grid levels remain almost the same in the original high resolution data and in the regridded data. In the correction process a constraint is put on the smoothness of the profile, thereby preventing the correction from producing strongly oscillating profiles. The results are regridded data consisting of reasonably smooth profiles having practically the same partial columns as the original high resolved radiosonde profiles. For the high altitudes that are not detected by the GRUAN radiosonde we use the retrievals a priori data ( $x_a$ ).
- The regridded GRUAN in situ profiles g may be smoothed according to the averaging kernels of the remote sensing retrieval. The regridded and smoothed GRUAN in situ profile  $\hat{g}$  is then comparable to the remote sensing profile, whereby:

$$\hat{g} = (A_{11} + A_{12})(g - x_a) + x_a.$$
 (13)

Here  $A_{11}$  is the H<sub>2</sub>O block of the averaging kernel matrix and  $A_{12}$  the block that describes the response of the retrieved H<sub>2</sub>O on atmospheric HDO (see Sect. 2.3) and the vector  $x_a$  is the a priori state vector. An example illustrating the effects of the regridding and the smoothing is given in Fig. 10.

We would like to note that by using Eq. (13) we assume that H<sub>2</sub>O and HDO variations are fully correlated. However, H<sub>2</sub>O and HDO do not vary fully in parallel, i.e. calculating  $\hat{g}$  according to Eq. (13) implies an uncertainty that can be estimated by the uncertainty covariance matrix  $S_{\hat{g}}$  according to (see also Sect. 4.3 of Barthlott et al., 2017):

$$\mathbf{S}_{\hat{\mathbf{g}}} = \mathbf{A}_{12} \mathbf{S}_{\mathbf{a},\delta \mathbf{D}} \mathbf{A}_{12}^{T}.$$
(14)

Here  $\mathbf{S}_{\mathbf{a},\delta D}$  describes the actual atmospheric  $\delta D$  covariances. Because  $\mathbf{A}_{12}$  and  $\mathbf{S}_{\mathbf{a},\delta D}$  have small entries only, this uncertainty is below 1% and can be neglected for our comparison.

### 5.2 Metric for quantifying data agreement

20

For a better statistical quantification of the deviations of the remote sensing data from the GRUAN reference data, we introduce a skill score DL describing the difference of the logarithmic values of the respective water vapour concentrations. Because
 30 Δln(x) ≈ Δx/x, we interpret the logarithmic scale difference between IASI and GRUAN as the relative difference (and use the GRUAN data in the denominator). DL then becomes:

$$DL = \ln ([H_2O]_{retrieval}) - \ln ([H_2O]_{GRUAN})$$
  

$$\approx \frac{[H_2O]_{retrieval} - [H_2O]_{GRUAN}}{[H_2O]_{GRUAN}},$$
(15)

where  $[H_2O]_{GRUAN}$  is the regridded and smoothed radiosonde  $H_2O$  data (i.e.  $\hat{g}$  from Eq. 13) and  $[H_2O]_{retrieval}$  is the retrieved IASI  $H_2O$  data. The so defined skill score DL is a good measure for the relative difference between the GRUAN and IASI data. As a good measure for the mean relative difference between GRUAN and IASI we can use the mean difference of logarithmic

values (MDL):

$$MDL = \frac{1}{N} \sum_{i=1}^{N} DL_{i} = \frac{1}{N} \sum_{i=1}^{N} \left[ \ln \left( [H_{2}O]_{retrieval} \right) - \ln \left( [H_{2}O]_{GRUAN} \right) \right]_{i}$$
$$\approx \frac{1}{N} \sum_{i=1}^{N} \left( \frac{[H_{2}O]_{retrieval} - [H_{2}O]_{GRUAN}}{[H_{2}O]_{GRUAN}} \right)_{i}.$$
(16)

Similarly, we can use the standard deviation of the logarithmic differences as a measure for the relative scatter between GRUAN and IASI and introduce  $\sigma_{MDL}$  as

$$\sigma_{\text{MDL}} = \sqrt{\frac{1}{N} \sum_{i=1}^{N} (\text{DL}_i - \text{MDL})^2}.$$
 (17)

For illustrating the variation of the atmospheric state we introduce  $\sigma_{\hat{g}}$  as

$$\sigma_{\hat{g}} = \sqrt{\frac{1}{N} \sum_{i=1}^{N} \left[ \ln\left( [\mathrm{H}_{2}\mathrm{O}]_{\mathrm{GRUAN}} \right)_{i} - \overline{\ln\left( [\mathrm{H}_{2}\mathrm{O}]_{\mathrm{GRUAN}} \right)} \right]^{2}}.$$
(18)

We want to document to what extent the differences between GRUAN and MUSICA IASI data can be explained by the estimated errors. In Sect. 4 we estimate in detail the error in the MUSICA IASI H<sub>2</sub>O profiles for three different climate zones.
Uncertainties in the GRUAN H<sub>2</sub>O profiles have also to be considered. In general the uncertainty of the GRUAN data increases with altitude. For the regridded and smoothed GRUAN profiles Δĝ is about 3-5% near the surface and 5-20% at around 10 km altitude. For further details on the radiosonde uncertainty we refer to Appendix A. If we assume that the MUSICA IASI and the GRUAN errors are uncorrelated random errors we can calculate the 1σ scatter of DL around MDL as expected from the MUSICA IASI and GRUAN errors by:

$$\Delta_{\text{MDL}} = \sqrt{\frac{1}{N} \sum_{i=1}^{N} \left( \Delta \hat{x}_i^2 + \Delta \hat{g}_i^2 \right)}.$$
(19)

Here the index i stands for an individual observation and N is the number of all observations.

### 5.3 Data agreement for individual ensembles

In this section we present the comparison between the regridded and smoothed GRUAN  $H_2O$  profiles and the IASI  $H_2O$  profiles using the metric as described in the previous subsection. The statistical quantifications are made individually for the

25 four different ensembles as given in Table 1. The aim is to illustrate the remote sensing data quality for the three different climate zones.

### 5.3.1 MUSICA IASI standard retrieval

Figure 11 depicts the vertical distribution of the data agreement for the MI and LI08 ensembles. These ensembles correspond to IASI measurements with available EUMETSAT L2 data and we can execute the standard MUSICA IASI retrieval, which

- uses the EUMETSAT L2 temperature data as the a priori atmospheric temperatures. For the MI ensemble the MDL value (thick black line) oscillates between -23% and -7% below 10 km altitude and is close to zero at higher altitudes. The scatter  $\sigma_{MDL}$  is indicated by the black error bars and it is generally within 20% except for the altitudes around 12 km where it is slightly higher. For the LI08 ensemble the MDL values oscillates between -20% and +16% and the scatter  $\sigma_{MDL}$  is up to 42% below 5 km and about 15% at higher altitudes. For both ensembles (MI and LI08) the  $\sigma_{MDL}$  values are significantly smaller than the
- 1 $\sigma$  variation in the smoothed radiosonde data ( $\sigma_{\hat{g}}$ ).

The red shaded area around the MDL value represents the  $\Delta_{MDL}$  values, i.e. the scatter in the MDL value we expect due to the errors in the MUSICA IASI and GRUAN H<sub>2</sub>O data. The  $\Delta_{MDL}$  values are calculated according to Eq. (19) by considering MUSICA IASI random errors due to measurement noise, emissivity and atmospheric temperature (actually we work with the error estimations as depicted in Fig. 7) and the GRUAN random errors as discussed in Appendix A and presented in Fig. A2.

- For the MI ensemble the  $\Delta_{MDL}$  and  $\sigma_{MDL}$  show similar amplitudes and vertical behavior, meaning that the expected and the observed scatter agree reasonably well. For the LI08 ensemble  $\Delta_{MDL}$  and  $\sigma_{MDL}$  agree well only above 5 km altitude. At lower altitudes the actually observed scatter is significantly larger than the scatter expected from the estimated MUSICA IASI and GRUAN errors, which might indicate an underestimation of the MUSICA IASI random errors at Lindenberg below 5 km altitude.
- The comparison suggests a weak dry bias in the MUSICA IASI data between 2 and 10 km at Manus Island and above 10 km at Lindenberg. The former could be explained by errors in the simulated line intensities and the latter by errors in the simulated line shapes (see discussion in the context of Fig. 8). However, given the small number of ensemble members we should be careful and avoid premature conclusions.

### 5.3.2 Retrieval using external temperature data

- During summer 2007 EUMETSAT L2 data have not been available and the retrievals for the LI07 and SK07 ensembles have been executed using the atmospheric temperature measured by the GRUAN radiosondes as the a priori atmospheric temperatures (see discussion in Sect. 3.4). In order to avoid inconsistencies when comparing the different ensembles we simulate retrievals of the MI and LI08 ensembles that also use the GRUAN radiosonde temperatures instead of the EU-METSAT L2 temperatures as the a priori atmospheric temperatures. The simulated retrieval products are obtained by adding
- $GK_T(T_{L2} T_{GRUAN})$  to the standard MUSICA IASI retrieval products, where G is the gain matrix,  $K_T$  the Jacobian matrix for atmospheric temperature, and  $T_{L2}$  and  $T_{GRUAN}$  are the atmospheric temperature state vectors of the EUMETSAT L2 and GRUAN data, respectively. For the altitudes above the radiosonde we extend the  $T_{GRUAN}$  vector with a zonally and monthly

mean temperature climatology (Rees et al., 1990). For calculating the combined MUSICA IASI and GRUAN random error (i.e. the expected scatter  $\Delta_{MDL}$ ) we have to consider the uncertainties in the GRUAN temperatures instead of the uncertainties in the EUMETSAT L2 temperatures. Appendix B gives a brief overview on the GRUAN temperature uncertainties.

Figure 12 depicts the data agreement for all four ensembles when GRUAN radiosonde temperatures are used as the a priori atmospheric temperatures. For Manus Island and Lindenberg (ensembles MI and LI08) the scatter in MDL ( $\sigma_{MDL}$ ) is signifi-

- cantly reduced if compared to Fig. 11 (figure showing the data agreement for MUSICA IASI retrievals that use EUMETSAT L2 temperatures as the a priori atmospheric temperature). A similar reduction is also observed in the theoretically predicted scatter ( $\Delta_{MDL}$ ), because the GRUAN temperatures have a much smaller uncertainty (typically 0.1-0.3 K) than the EUMETSAT L2 temperatures (we assume 1-2 K). At Lindenberg (ensemble LI08)  $\sigma_{MDL}$  and  $\Delta_{MDL}$  agree much better for the retrieval products obtained by using GRUAN temperature as a priori than for the MUSICA IASI standard retrieval products (obtained by using
- EUMETSAT L2 temperature as a priori). This suggests that for Lindenberg and the year 2008 our uncertainty assumptions for the EUMETSAT L2 atmospheric temperatures (see Table 2) are probably too optimistic.

The bottom panels in Fig. 12 show the data agreement for the LI07 and SK07 ensembles. These ensembles are exclusively representative for summer observations. We observe that the  $\sigma_{MDL}$  values are generally larger than the  $\Delta_{MDL}$  values, meaning that we probably underestimate the MUSICA IASI random errors. In addition we find a wet bias of up to 30% below 2 km altitude and a dry bias of about 20% at around 14 km.

An upper tropospheric dry bias is consistently observed in the analysis of the LI08, LI07 and SK07 ensembles, but not seen in the analysis of the MI ensemble. A systematic uncertainty source that affects upper tropospheric  $H_2O$  at Lindenberg and Sodankylä but not at Manus Island is the shape of the water vapour lines (see discussion in the context of Fig. 8). So deficits in simulating the line shapes might explain this upper tropospheric dry bias. In the near surface atmosphere we observe a

- wet bias at the two continental sites Lindenberg and Sodankylä, but only for the ensembles that are limited to the summer season (LI07 and SK07). Our error estimation study suggests that small uncertainties in the emissivity can cause large errors at these continental sites. So an uncertainty in the used IREMIS emissivity is a candidate for explaining the surface near wet bias; however, the H<sub>2</sub>O retrieval response for a -1 % uncertainty in the emissivity differs betweeen observations and can be positive or negative (see Fig. 9). This means that emissivity uncertainties can only explain the bias if the sign of the emissivity
- uncertainty is correlated with the atmospheric state (e.g. the uncertainty in the used monthly IREMIS surface emissivity is typically positive for dry atmospheric conditions and typically negative for humid atmospheric conditions) or surface conditions (e.g. the uncertainty in the IREMIS data is typically positive/negative for a surface with high/low emissivity or high/low skin temperatures).

### 5.4 Global overview on data agreement

15

30 Figure 13 depicts the vertical profiles of the data agreement skill score parameters for all coinciding observations without separating the different ensembles. This analysis is based on 100 individual comparisons.

Below 12 km altitude the MDL value oscillates between -10% and +11% and at around 14 km altitude it reaches -21%. The scatter in MDL ( $\sigma_{MDL}$ ) is almost 29% close to the surface but generally smaller than 20% above 1 km altitude. Above 5 km altitude this observed scatter is only slightly larger than the scatter predicted from the estimated errors ( $\Delta_{MDL}$ ). At lower altitudes the predicted scatter is clearly smaller than the observed scatter. An explanation of the observed upper tropospheric bias and increased scatter at low altitudes is given in the previous section: the dry bias in the upper troposphere might have its origin in an incorrect modeling of the spectroscopic line shapes and the increased scatter near the surface might be due to uncertainties in the IREMIS emissivities.

- The observed scatter between GRUAN and IASI ( $\sigma_{MDL}$ ) is significantly smaller than the  $1\sigma$  variation of the smoothed radiosonde data ( $\sigma_{\hat{g}}$ ), which reaches about 50% near the surface and more than 100% in the middle and upper troposphere. This reflects the large variation in the atmospheric water vapour concentration data we use for our evaluation study (see also Fig. 2). The MUSICA IASI data product does well capture most of these variations. For demonstrating this capability Fig. 14 illustrates correlations between the MUSICA IASI retrieval products and the smoothed GRUAN data for selected altitudes. The
- respective altitudes are highlighted in Figs. 3 and 5, which documents that at all sites the MUSICA IASI product for 1.8, 3.6, 6.4, and 9.8 km are independent and well sensitive to real atmospheric variations. Near the surface the sensitivity is generally limited and at 13.6 km only the Manus Island data are reasonably sensitive to the actual atmospheric variations.

At the altitudes where the MUSICA IASI product show very good sensitivity (1.8, 3.6, 6.4, and 9.8 km) we observe a very good correlation and can demonstrate that the MUSICA IASI product can correctly capture the large variations that are present

- in atmospheric water vapour. For instance, at 3.6 km the mixing ratios range from below 200 ppmv to almost 20000 ppmv and at 9.8 km from 10 ppmv to more than 1000 ppmv. Please note that these large variations are reliably reproduced by the MUSICA IASI processor, although the retrieval works with a single humidity a priori value that is used at all sites and during all seasons (indicated as the yellow star in Fig. 14). Near the surface the correlation is a bit weaker than at higher altitudes, mainly due to the some outliers belonging to the LI07 and SK07 ensembles (the ensembles representing the summer season
- over land). At 13.6 km altitude we observe a good correlation, which demonstrates the possibility of detecting  $H_2O$  at Manus Island. However, at Lindenberg and Sodankylä these variations are strongly driven by actual atmospheric variations that take place at lower altitudes (see magenta lines in Figs. 3 and 5).

For our theoretical error analyses in Sect. 4 we assume that the relative errors have a component that is mostly random (Fig.7) and another component that is mostly systematic (Fig.8). For the comparison study we proceed similarly and examine bias and scatter, which means that we describe the variances in the MUSICA IASI data by the variances in the GRUAN data  $(\sigma_{\hat{g}}^2)$  and the variance in the difference between MUSICA IASI and GRUAN ( $\sigma_{\text{MDL}}^2$ ). Using this description we can calculate the  $R^2$  value that represents the portion of the MUSICA IASI variance that is in full agreement (fully correlated) with the GRUAN variance:

$$R^2 = \frac{\sigma_{\hat{g}}^2}{\sigma_{\hat{g}}^2 + \sigma_{\text{MDL}}^2}.$$
(20)

Each panel of Fig. 14 contains the  $R^2$  value calculated for the respective altitude.

The error blue bars on the diagonal of the plots of Fig. 14 indicate the typical GRUAN errors ( $\Delta \hat{g}$  as detailed in Appendix A) and the root-square-sum of the typical leading MUSICA IASI random errors ( $\Delta x$ ), whereby we have considered measurement

noise, uncertainties in surface emissivity and uncertainties in the GRUAN temperatures. The MDL and  $\sigma_{MDL}$  values are also written in each panel as bias (b) and scatter (s) value, respectively. They are the same as shown in Fig. 13.

Figure 15 resumes the capability of the MUSICA IASI retrieval product for capturing real atmospheric H<sub>2</sub>O variations at different altitudes by showing vertical profiles of the  $R^2$  values calculated according to Eq. (20) for the different ensembles individually and when considering all 100 individual comparisons together. Between 1 km and 12.5 km altitude (and when

considering all comparisons together the MUSICA IASI products) detects more than 90% of the atmospheric variations in agreement with GRUAN.

### 6 Summary and outlook

In this paper, we compare water vapour profiles retrieved from IASI spectra by the MUSICA IASI retrieval with in situ measurements from GRUAN radiosondes at three different reference sites representative of three different climate zones (tropics, mid-latitudes and polar region). In addition, we provide an extensive theoretical error estimation of the retrieval's water vapour product for the respective reference sites considering many different uncertainty sources.

The error estimations of the MUSICA IASI water vapour profiles at the different reference sites reveal that for the lowermost 3 km the errors can be as large as 30%. The most important uncertainty sources are unrecognized clouds, and uncertainties in lower tropospheric temperature and in surface emissivity. Between 3 and 6 km the error can be as large as 20%, mainly due to

15 middle atmospheric temperature uncertainties and unrecognized high cirrus clouds. Above 6 km the errors are typically smaller than 20% and mainly caused by uncertainties in upper tropospheric temperatures and uncertainties in spectroscopic pressure broadening parameters.

For the empirical validation study the remote sensing MUSICA IASI  $H_2O$  profiles have been compared to 100 different Vaisala RS92 radiosonde measurements that have been processed by the GRUAN lead centre. The scatter found for the difference between GRUAN and IASI is smaller than 21% above 1.8km altitude. It is slightly higher near the ground. This is in good

- agreement with errors as given for the GRUAN data and the errors as estimated for the MUSICA IASI product. It is important to note that the coincidences correspond to five different years and represent three different climate zones, giving the study here presented a good global representativeness. We demonstrate that the MUSICA IASI retrieval is able to correctly capture variations in  $H_2O$  profiles between 1 km above ground up to the upper troposphere.
- The comparison indicates to a dry bias of the remote sensing data of 20% in the upper troposphere of the middle and high latitude sites, but not at the tropical site. We find that deficits in spectroscopic line shape modeling could explain such behavior. For the current MUSICA IASI retrieval a Voigt line shape model is assumed and HITRAN 2016 pressure broadening parameters are used. It would be interesting to investigate if the usage of more sophisticated line shape models (e.g. speeddependent Voigt line shape model) could reduce the upper tropospheric bias and improve the agreement between the MUSICA
- 30 IASI remote sensing and GRUAN in-situ data. For the continental sites (Lindenberg and Sodankylä) and during summer we observe a wet bias in the MUSICA IASI data with respect to GRUAN. Uncertainties in land surface emissivity being correlated to atmospheric or surface conditions (e.g. negative/positive emissivity uncertainties occurring in line with very dry/humid

atmospheric conditions or hot/cold skin temperatures) could explain this behavior. It would be interesting to test if the usage of a daily surface emissivity product instead of the monthly mean IREMIS data (which have been used for the here presented retrievals) improves the agreement between MUSICA IASI and GRUAN.

## 7 Data availability

The MUSICA IASI data presented here are available on the MUSICA website http://www.imk-asf.kit.edu/english/musica.php.

5 Please contact M. Schneider for more details. The GRUAN data are available at the GRUAN website: https://www.gruan.org/ data/data-products/gdp/rs92-gdp-2/.

### Appendix A: Uncertainties of GRUAN water vapour volume mixing ratios

In order to perform a valid comparison between remote sensing data and in situ measurements, the uncertainties of the in situ data have to be considered.

GRUAN provides uncertainties for the relative humidity ( $\Delta \rho$ ), for the temperature ( $\Delta T$ ), and for the pressure ( $\Delta p$ ). The 5 water vapour volume mixing ratio (WVMR) is defined as

WVMR = 
$$\frac{\varrho E(T)}{p - \varrho E(T)} \approx \frac{\varrho E(T)}{p}$$
, (A1)

where E is the water vapour saturation pressure. The GRUAN WVMR error for each individual radiosonde can be calculated as

WVMR<sub>e</sub> = 
$$\sqrt{\left(\frac{\Delta E(T)}{E(T)}\right)^2 + \left(\frac{\Delta \varrho}{\varrho}\right)^2} \times WVMR.$$
 (A2)

10 Uncertainties in atmospheric pressure p can be neglected if compared to the uncertainties of E(T) and  $\rho$ . For the calculation of the water vapour saturation pressure we use the same formula as GRUAN from Hyland and Wexler (1983). Since E(T) is a highly non-linear function, we estimate the uncertainty of E by

$$\Delta E = \max\{|E(T + \Delta T) - E(T)|; |E(T - \Delta T) - E(T)|\}.$$
(A3)

According to Dirksen et al. (2014) there are correlated and uncorrelated errors. We investigate both separately. Figure A1
depicts the correlated and uncorrelated GRUAN WVMR errors (WVMR<sub>e</sub>) in the top and bottom panels, respectively. Black lines indicate the data ensembles that cover all seasons (Manus Island and Lindenberg 2008) and red lines the ensembles that are only representative for the summer season (Lindenberg 2007 and Sodankylä).

- For a reasonable comparison the vertically highly resolved GRUAN profiles have to be adjusted to the vertical resolution of the remote sensing profiles (see Sect. 5.1). This means a significant reduction of the vertical resolution and the uncorrelated errors will cancel out. The regridding and smoothing of the correlated errors is accomplished as follows: First, the errors  $WVMR_e$  are added to the measured WVMR data. Second, for  $WVMR + WVMR_e$  we perform the regridding as described in Sect. 5.1, i.e. we calculate the regridded version of the erroneous GRUAN WVMR profile. The difference between the erroneous and the original profiles (of the regridded versions) give the regridded GRUAN WVMR uncertainty profile ( $\Delta g$ ). Above the radiosonde (where we set g equal to the retrievals a priori) we set the uncertainty to 100%. Then we calculate an
- 25 uncertainty covariance matrix  $S_g$  using the values from the uncertainty profile  $\Delta g$  and a large correlation length of 30 km individually for the two blocks representing the data measured by the radiosonde and the data above the radiosonde. Third, in analogy to Eq. (13) we apply the averaging kernels to  $S_g$  and get the error covariance for the regridded and smoothed GRUAN profiles as:

$$\mathbf{S}_{\hat{\mathbf{g}}} = (\mathbf{A}_{11} + \mathbf{A}_{12})\mathbf{S}_{\mathbf{g}}(\mathbf{A}_{11} + \mathbf{A}_{12})^{T}.$$
(A4)

30 Figure A2 depicts the square root values of the diagonal of  $S_{\hat{g}}$  for the different ensembles. The uncertainties typically increase from 5% near the ground to 5-20% at around 10 km altitude. For higher altitudes it decreases again due to the decaying sensitivity (see averaging kernel plots of Fig. 3).

### **Appendix B: Uncertainties of GRUAN temperatures**

The MUSICA IASI retrival uses the EUMETSAT L2 temperatures as the a priori atmospheric temperatures. However, in summer 2007 EUMETSAT L2 data are not available and instead we use the GRUAN temperatures for the LI07 and SK07 retrievals. In addition, for the MI and LI08 ensembles we simulate retrievals that use the GRUAN temperatures instead of the

5 EUMETSAT L2 temperatures as the a priori atmospheric temperatures. For all these retrievals the uncertainty in the GRUAN temperatures and not the uncertainty in the EUMETSAT L2 temperatures has to be considered for the error estimation.

Figure A3 depicts the correlated GRUAN temperature uncertainty profiles after regridding the data to the MUSICA retrieval grid points by using a triangle inverse-distance weighted averaging function (as for the first  $H_2O$  regridding step, see Sect. 5.1). We assume that the uncorrelated uncertainties cancel out by this averaging. Below 20 km altitude the GRUAN temperature

- uncertainties are well within 0.3 K, i.e. they are much smaller than the uncertainties of 1-2 K we assume for the EUMETSAT L2 temperatures. The GRUAN nighttime temperature data have an uncertainty that is rather constant with altitude, whereas for the daytime data the uncertainty monotonically increases with altitude. Above the top altitude of the radiosonde we use a monthly and zonally averaged temperature climatology (COSPAR International Reference Atmosphere; Rees et al., 1990) and assume a temperature uncertainty of 5 K (this explains the instantaneous uncertainty increase that can be observed for some
- Manus Island and Lindenberg radiosondes).

*Author contributions.* C. Borger performed most calculations for this work during his master thesis at KIT IMK-ASF and prepared the manuscript together with M. Schneider and in collaboration with all coauthors. M. Schneider developed the IASI retrievals in the framework of the MUSICA project and B. Ertl supported these developments by making the processing chain more efficiently. F. Hase wrote the PROFFIT and PRFFWD codes. O. García helped in reading and formatting the EUMETSAT IASI L2 data. M. Sommer provided the GRUAN

- radiosonde measurements in a very useful data format for the sites of Lindenberg and Sodankylä. M. Höpfner helped us with the KOPRA calculations used for estimating the effect of the scattering by cirrus and mineral dust particles. S. Tjemkes provided all necessary data for the site of Manus Island in a very useful data format in the framework of a planned exercise called "Intercomparison of Hyperspectral Retrieval Codes". X. Calbet collected the IASI/GRUAN coincidences over Manus Island and helped us in the interpretation of the radiosonde's measurement uncertainties.
- Acknowledgements. This research has largely benefit from results of the projects MUSICA (funded by the European Research Council under the European Community's Seventh Framework Programme (FP7/2007-2013) / ERC Grant agreement number 256961), MOTIV (funded by the Deutsche Forschungsgemeinchaft under GZ SCHN 1126/2-1) and INMENSE (funded by the Ministerio de Economía y Competividad from Spain, CGL2016-80688-P). We acknowledge the support by the Deutsche Forschungsgemeinschaft and the Open Access Publishing Fund of the Karlsruhe Institute of Technology.

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

**Figure 1.** Example of an infrared spectrum measured by IASI (upper panel) and residuals between satellite observation and radiative transfer simulation (bottom panel) at Manus Island (2012-10-15 11:46:26 UT, satellite zenith angle 10.2°, integrated water vapour 48.0 mm). The red lines in the bottom panel indicate the typical IASI noise measurement level as given by the square root values of the diagonal elements of the IASI noise covariance matrix (Pequignot et al., 2008).