# Peer review of "Evaluation of MUSICA IASI tropospheric water vapour profiles by theoretical error assessments and comparisons to GRUAN Vaisala RS92 measurements"

_Atmospheric Measurement Techniques, 2017_

## Referee Comment (RC1) · Anonymous Referee #1 · 17 Mar 2018

General Comments

This manuscript presents IASI water vapour VMR profiles retrieved using the MUSICA algorithm and evaluates them against coincident GRUAN radiosonde profiles measured at three stations representing tropical, mid-latitude, and high-latitude conditions. The MUSICA MetOp/IASI water vapour retrieval method is described, averaging kernels and DOFS are presented, and the error budget terms are thoroughly evaluated. Vertical profiles of the errors are used to identify the dominant sources of uncertainty as a function of altitude. A total of 100 coincident GRUAN profiles are used for validation.

[Figure]

These are regridded to the IASI altitude grid and smoothed with the averaging kernels prior to the comparisons. Correlation plots at three altitudes and vertical profiles of the mean relative difference and its standard deviation are used to quantify the agreement between the two datasets. Overall, IASI and GRUAN differ by less than 25% between 1.5 and 10 km altitude, and are within the errors for the two data products. The authors conclude that the MUSICA MetOp/IASI retrieval processor provides water vapour VMR profiles with good accuracy and captures variations between 1.5 km above ground up to the tropopause with a precision that is consistent with the theoretical error assessment.

The manuscript is clearly and concisely written and I recommend publication after the minor comments below are addressed.

Specific Comments

Page 4, lines 9-15 – Se is used to denote the error covariance matrix in both Eqs. 8 and 9, implying that these are the same. However, the former is due to parameter uncertainties and the latter is due to measurement uncertainty. Rename one of them to clearly differentiate?

Page 6, line 12 – 100 coincident IASI-GRUAN profiles is not a large number for a validation study, particularly for such a highly variable species as water vapour, even though they do cover three representative sites. Some discussion to justify that this number sufficient for good statistics should be added, perhaps citing other water vapour validation studies.

Page 7, line 5 – Provide the collocation criteria for the Manus Island comparisons, as done for Lindenberg (lines 14-15). Similarly, provide them for Sodanklya at line 22.

Page 7, lines 17-19 – Shouldn't this information about the Manus Island IASI data version be included in Section 3.2? Comment on whether there are differences between v4 and v5. Also in Section 3.4, state which version is used for Sodankyla.

**AMTD**

Page 10, line 12 – The text says that "The right panel of Fig. 5 illustrates that . . . a cirrus cloud has a weaker dependency on wavenumber than a dust layer." However, in this figure, the line for cirrus cloud (green) decreases more with wavenumber than does the line for dust (blue), suggesting that cirrus has a stronger dependency on wavenumber. Clarify.

Page 11, line 4 – Here and elsewhere, why use "error pattern profile" rather than "error profile"? The xe in Eq. 7 and plotted in Figs. 7, 8, and 9, is just defined as the error. Delete "pattern" throughout, or define it.

Page 11, lines 15-20 – This paragraph doesn't comment on the oscillations seen in the lower panels of Fig. 7. Add some discussion.

Page 14, Section 5.1 and Figure 11 – In the text or the figure, provide the slopes in linear space and the correlation coefficient R2 for each of the panels.

Page 14, line 18-19 – Explain why the two terms in Eq. 15 are approximately equivalent.

Technical Corrections

Page 1, line 8 – degreeS

Page 1, line 13 – coincidENT

Page 1, line 15 – but never exceeds 30%

Page 1, line 17 – in accordance WITH the

Page 2, line 10 – aircraft AND satellites

Page 2, line 16 – delete respective

Page 2, line 34 – outcomeS

Page 3, line 3 – introduction to the

Page 3, line 8 – THE forward model F), which relates the

Page 3, line 9 – add period after equation

Page 3, line 18 – delete level

Page 3, line 20 – whereby THE kind

Page 4, line 21-22 – reorder cited references

Page 4, line 22 – delete ranging

Page 4, line 27 – HITRAN 2016? could state this explicitly

Page 5, line 3 – a priori PROFILE

Page 5, line 11 – by THE US Geological Survey

Page 5, line 17 – differences COMPARED to the

Page 6, line 11 – observationS

Page 6, line 16 – could change "mid-latitudinal" to "mid-latitude" throughout

Page 6, line 20 – FOR more details

Page 7, line 5 and 13– coincidences WITH

Page 7, line 13 – representative OF all

Page 7, line 16 – As for Manus

Page 8, line 14 – Fig. 4 (not 5)

Page 9, line 7 – delete hypothetically

Page 9, line 13 – coverage BY opaque

Page 10, line 24 – The error profiles shown are

Page 11, line 32 – maximUM errors

Page 12, line 2 – in Fig. 8.

Page 12, line 4 – Strictly, Fig. 9 shows the influence of cloud type on the errors, rather than the retrieval.

Page 12, line 5 – Now using "humidity" as equivalent to water vapour VMR – is that correct?

Page 12, line 10 – Fig. 9 middle row shows results for clouds at 4.9 km, not 3.0 km, at Manus Island and Lindenberg. Clarify this in the discussion.

Page 12, line 10 – Elsewhere, altitude above ground is referenced, rather than altitude above mean sea level. Check that terminology is correct.

Page 12, line 13-14 – errors are more than 10%.

Page 12, line 18 – (FOR details

Page 12, line 21 – This differs from the

Page 12, line 22 – for THESE different

Page 12, line 25 – "So we have to" – rather colloquial

Page 12, line 31 – put ON the . . .

Page 13, line 1 – thereby preventing the correction from producing strongly oscillating profiles

Page 13, line 32 – and in situ data identify well

Page 14, line 8 – delete if

Page 14, line 9 – VMRs (rather than concentrations – two references)

Page 15, line 5 – agreement WITH the

Page 15, line 14 – representative OF three

[Figure]

Page 15, line 28 – giving the study presented here a good

Page 16, line 2 – The MUSICA MetOp/IASI data presented here are

Page 17, line 7 – radiosonde CAN be

Page 17, line 14 – depicts the correlated

Page 17, line 26 – ground to 5-20%

Page 22, Figure 1 – spectrUM

Page 25, Figure 4 – degreeS; spelling of Lindenberg is incorrect in the figure

Page 26, Figure 5 – Since (4-6km) altitude range is given for Dust in the legend, could add (13-14km) for Cirrus.

Page 27, Figure 6 – error profiles derived from instrument noise

Page 28, Figure 7 – delete different

Page 29, Figure 8 – Font size for site names is different from Figures 7 and 9. Also smaller in Figure A.2.

Page 31, Figure 10 – x-axis label is Water vapour, but H2O used in other figures. In caption: MetOp/IASI retrieval OF H2O profiles.

Page 32, Figure 11 – Add linear slopes and correlation coefficient R2 to each panel. In caption: Red and black colourS

---

## Short Comment (SC1) · 4 Apr 2018

General Comments

This manuscript presents IASI water vapour VMR profiles retrieved using the MUSICA algorithm and evaluates them against coincident GRUAN radiosonde profiles measured at three stations representing tropical, mid-latitude, and high-latitude conditions. The MUSICA MetOp/IASI water vapour retrieval method is described, averaging kernels and DOFS are presented, and the error budget terms are thoroughly evaluated. Vertical profiles of the errors are used to identify the dominant sources of uncertainty as a function of altitude. A total of 100 coincident GRUAN profiles are used for validation. These are regridded to the IASI altitude grid and smoothed with the averaging kernels prior to the comparisons. Correlation plots at three altitudes and vertical profiles of the mean relative difference and its standard deviation are used to quantify the agreement between the two datasets. Overall, IASI and GRUAN differ by less than 25% between 1.5 and 10 km altitude, and are within the errors for the two data products. The authors conclude that the MUSICA MetOp/IASI retrieval processor provides water vapour VMR profiles with good accuracy and captures variations between 1.5 km above ground up to the tropopause with a precision that is consistent with the theoretical error assessment. The manuscript is clearly and concisely written and I recommend publication after the minor comments below are addressed.

We thank anonymous referee #1 for the very useful comments with respect to content, but also for the careful technical corrections.

Specific Comments

Page 4, lines 9-15 – Se is used to denote the error covariance matrix in both Eqs. 8 and 9, implying that these are the same. However, the former is due to parameter uncertainties and the latter is due to measurement uncertainty. Rename one of them to clearly differentiate?

We will make the notations in line with the recommendations that are currently elaborated by the TUNER team.
The error covariance matrix due to parameter uncertainties will be called: $\mathbf{S}_{x,b}$
The error covariance matrix due to noise in the measured spectra will be called $\mathbf{S}_{x,noise}$
Furthermore, we will use $b$ as the variable for parameters. The errors in $b$ will be described by $\Delta b$.
Errors in the retrieval state vector will be described as $\Delta x$. Equations (7) – (9) will be written as:
$$\Delta x = -\mathbf{G}\mathbf{K}_b \Delta b \qquad (7)$$
$$\mathbf{S}_{x,b} = -\mathbf{G}\mathbf{K}_b \mathbf{S}_b \mathbf{K}_b^T \mathbf{G}^T \qquad (8)$$
$$\mathbf{S}_{x,noise} = \mathbf{G}\mathbf{S}_{y,noise} \mathbf{G}^T \qquad (9)$$
Equations (1), (2), (3) and (4) will be changed accordingly.

Page 6, line 12 – 100 coincident IASI-GRUAN profiles is not a large number for a validation study, particularly for such a highly variable species as water vapour, even though they do cover three representative sites. Some discussion to justify that this number sufficient for good statistics should be added, perhaps citing other water vapour validation studies.

We agree that it would be desirable to have more profiles that can be compared. However, at the moment the here used 100 profiles are the only radiosonde profile data that have been processed with the GRUAN recommendations and that have been measured in coincidence with IASI observations. We will remark this in the manuscript and also discuss the use IASI-GRUAN collocation criteria in the context to other water vapour profile comparison studies.

Page 7, line 5 – Provide the collocation criteria for the Manus Island comparisons, as done for Lindenberg (lines 14-15). Similarly, provide them for Sodanklya at line 22.

Yes, we will better describe the collocation criteria for all three sites.

Page 7, lines 17-19 – Shouldn't this information about the Manus Island IASI data version be included in Section 3.2? Comment on whether there are differences between v4 and v5. Also in Section 3.4, state which version is used for Sodankyla.

Yes, for all sites and comparison periods we will describe the used IASI L2 PPF versions.

Page 10, line 12 – The text says that "The right panel of Fig. 5 illustrates that . . . a cirrus cloud has a weaker dependency on wavenumber than a dust layer." However, in this figure, the line for cirrus cloud

(green) decreases more with wavenumber than does the line for dust (blue), suggesting that cirrus has a stronger dependency on wavenumber. Clarify.

We have used the term "weaker", because the cirrus cloud has almost no signal at 1400 cm$^{-1}$, but a signal of -1.0 mWm$^2$sr$^{-1}$(cm$^{-1}$)$^{-1}$ at 1190 cm$^{-1}$. The dust cloud has a signal at 1400 cm$^{-1}$ of about -0.5 mWm$^2$sr$^{-1}$(cm$^{-1}$)$^{-1}$ at 1400 cm$^{-1}$. In order to avoid confusion we suggest using "another", instead of "a weaker".

Page 11, line 4 – Here and elsewhere, why use "error pattern profile" rather than "error profile"? The xe in Eq. 7 and plotted in Figs. 7, 8, and 9, is just defined as the error. Delete "pattern" throughout, or define it.

Ok, we agree. We will use error profile and delete "pattern".

Page 11, lines 15-20 – This paragraph doesn't comment on the oscillations seen in the lower panels of Fig. 7. Add some discussion.

The error in the atmospheric temperature has the strongest impact on the retrieved state vector x at the altitudes where the temperature error is located. This explains why the black, red, green and blue lines peak at different altitudes. The oscillation of a single error profile can be understood by Eq. (7), which is used for calculating the error profiles. Errors in the temperature propagate to the spectra according to **K**$_b$ and are then interpreted according to **G**, which in turn depends on the constraint (second part of the Cost function 2).

Page 14, Section 5.1 and Figure 11 – In the text or the figure, provide the slopes in linear space and the correlation coefficient R2 for each of the panels.

We will provide a Table with the R$^2$ values and the 95% confidence interval of the linear regression line slopes.

Page 14, line 18-19 – Explain why the two terms in Eq. 15 are approximately equivalent.

Because $\Delta \ln(x) = \frac{\Delta x}{x}$ we interpret the logarithmic scale difference between IASI and GRUAN as the relative difference (and use GRUAN data in the denominator).

Technical Corrections

We will consider all the technical corrections listed below.

Page 1, line 8 – degreeS

Page 1, line 13 – coincidENT

Page 1, line 15 – but never exceeds 30%

Page 1, line 17 – in accordance WITH the

Page 2, line 10 – aircraft AND satellites

Page 2, line 16 – delete respective

Page 2, line 34 – outcomeS

Page 3, line 3 – introduction to the

Page 3, line 8 – THE forward model F), which relates the

Page 3, line 9 – add period after equation

Page 3, line 18 – delete level

Page 3, line 20 – whereby THE kind

Page 4, line 21-22 – reorder cited references

Page 4, line 22 – delete ranging

Page 4, line 27 – HITRAN 2016? could state this explicitly

Page 5, line 3 – a priori PROFILE

Page 5, line 11 – by THE US Geological Survey

Page 5, line 17 – differences COMPARED to the

Page 6, line 11 – observationS

Page 6, line 16 – could change "mid-latitudinal" to "mid-latitude" throughout

Page 6, line 20 – FOR more details

Page 7, line 5 and 13– coincidences WITH

Page 7, line 13 – representative OF all

Page 7, line 16 – As for Manus

Page 8, line 14 – Fig. 4 (not 5)

Page 9, line 7 – delete hypothetically

Page 9, line 13 – coverage BY opaque

Page 10, line 24 – The error profiles shown are

Page 11, line 32 – maximUM errors

Page 12, line 2 – in Fig. 8.

Page 12, line 4 – Strictly, Fig. 9 shows the influence of cloud type on the errors, rather than the retrieval.

Page 12, line 5 – Now using "humidity" as equivalent to water vapour VMR – is that correct?

Page 12, line 10 – Fig. 9 middle row shows results for clouds at 4.9 km, not 3.0 km, at Manus Island and Lindenberg. Clarify this in the discussion.

Page 12, line 10 – Elsewhere, altitude above ground is referenced, rather than altitude above mean sea level. Check that terminology is correct.

Page 12, line 13-14 – errors are more than 10%.

Page 12, line 18 – (FOR details

Page 12, line 21 – This differs from the

Page 12, line 22 – for THESE different

Page 12, line 25 – "So we have to" – rather colloquial

Page 12, line 31 – put ON the . . .

Page 13, line 1 – thereby preventing the correction from producing strongly oscillating profiles

Page 13, line 32 – and in situ data identify well

Page 14, line 8 – delete if

Page 14, line 9 – VMRs (rather than concentrations – two references)

Page 15, line 5 – agreement WITH the

Page 15, line 14 – representative OF three

Page 15, line 28 – giving the study presented here a good

Page 16, line 2 – The MUSICA MetOp/IASI data presented here are

Page 17, line 7 – radiosonde CAN be

Page 17, line 14 – depicts the correlated

Page 17, line 26 – ground to 5-20%

Page 22, Figure 1 – spectrUM

Page 25, Figure 4 – degreeS; spelling of Lindenberg is incorrect in the figure

Page 26, Figure 5 – Since (4-6km) altitude range is given for Dust in the legend, could add (13-14km) for Cirrus.

Page 27, Figure 6 – error profiles derived from instrument noise

Page 28, Figure 7 – delete different

Page 29, Figure 8 – Font size for site names is different from Figures 7 and 9. Also smaller in Figure A.2.

Page 31, Figure 10 – x-axis label is Water vapour, but H2O used in other figures. In caption: MetOp/IASI retrieval OF H2O profiles.

Page 32, Figure 11 – Add linear slopes and correlation coefficient R2 to each panel. In caption: Red and black colourS

---

## Referee Comment (RC2) · Anonymous Referee #2 · 9 Apr 2018

This paper describes in detail an evaluation and error analysis of a water vapour profile retrieval algorithm (Multi-platform remote Sensing of Isotopologues for investigating the Cycle of Atmospheric water, MUSICA) that has been applied to the Infrared Atmospheric Soundings Interferometer (IASI) sensor flown onboard the EUMETSAT MetOp satellites. The algorithm is an optimal estimation (OE) algorithm after Rodgers (1990). The paper is well organized, thorough, precise and well written. As such, I recommend publication, but only after my comments/suggestions are addressed below:

**General Comments**

[Figure]

1. The Authors have neglected to reference any of the previous work regarding satellite sounder validation as detailed in numerous publications, especially those pertaining to the NASA Atmospheric Infrared Sounder (AIRS). In fact, I don't even recall that the AIRS sounder was even given a mention in this paper. As the Authors must know, acknowledgement of previous related work is a very important aspect of science publications. I can provide example publications, but I'm sure the Authors are already aware of them. It is important that this previous work be acknowledged in the revision.

2. On Page 7 the Authors indicate that they are using Level 2 IASI products as components in their a priori state. Unless the Authors are using independent channels from those used in the EUMETSAT L2 product, then this technically speaking would not be an a priori, which by definition is a "virtual measurement" not used prior to the retrieval. Also, this begs the question: Why not simply use the L2 $H_2O$ product? The Authors should acknowledge and clarify these considerations, and it is particularly important that they explain the latter question — else the paper's novelty and/or utility comes into question.

**Specific Comments**

- Page 2, Line 26: "MUSICA" — acronym should also be defined here.

- Section 2.1: Nicely written introduction to the OE methodology — this is greatly appreciated.

- Page 3, Lines 3–4: This statement needs to be qualified with adequate references. As the authors probably know, the NASA AIRS sounding algorithm (which is the pathfinder high spectral resolution infrared sounder) does not use the Rogers version of OE.

- Page 3, Equation (2): recommend that this be expressed as a formal equation — the "cost" term can be explicitly featured on the left side (a standard variable is $J$, but the authors are free to use whichever they choose).

- Page 4, Lines 19–20: Acronym should be defined above at first occurrence.

- Page 4, Line 24: "HDO" — this chemical formula is not something I've encountered — although I (and astute readers) may deduce that it has something to do with the isotope formulas, the Authors should nevertheless define it and describe why it's relevant and/or important to this paper.

- Page 5, Lines 2–3: By "single a priori" do you mean a single a priori profile globally? If so, where do you obtain this?

- Page 5, Line 5: "there is no constraint on the surface temperature" — What is meant by this? What is the a priori for surface temperature?

- Page 5, Lines 14–15: Although not critical to the current paper, the Authors may wish to consider more recent models in future work, such as Watts et al. (1996), Masuda (2006), or the JCSDA CRTM model (Nalli et al., 2008), and perhaps acknowledge this in the text. The Masuda et al. (1988) model is known to have significant biases at larger scan angles. Also, what is being used as the surface wind speed?

- Page 6, Lines 19–20: The meaning of this sentence is not clear — I'm not sure how different time periods at the different locations means that the the dataset is not uniform.

- Page 6, Section 3.1: What is the source for the Vaisala sensor information? This should also be included in the references.

- Page 7, Line 5: I was not aware that there was a GRUAN site at Manus Island. When did this come online and is it still in operation?

- Page 7, Lines 8–9: See General Comment #2 above.

- Page 7, Lines 13–15: Why is this detail not given for the Manus site?

- Page 7, Lines 24–25: I'm not sure I fully understand — if the radiosondes are being used as truth, then how can they be used as the a priori?

- Page 8, Line 5: "this altitudes" should be "these altitudes"

- Page 12, Line 25: "about every 10 m" — I'm not sure this is correct. My understanding is that the balloon ascent is about 5 m/s, and the Vaisala processed radiosonde reports every second.

- Page 12, Line 27: Insert "statistically" between "performed" and "in two steps", thus "is performed statistically in two steps"

- Page 13, Lines 3–4: Replace "In order to get the in situ profile data that are comparable to the remote sensing data we have to smooth the" simply with "The" and insert "may be smoothed" after $x_{\mathrm{GRUAN}}$ and "according"

- Page 14, Line 17: The logarithmic dependence of these formulations is not immediately apparent in the second lines of these equations, which simply denote relative values. I recommend inserting the intermediary mathematical step.

- Page 15, Line 4: Recommend using percents here so that it is clear that one is talking about relative values.

- Page 17, Line 7: "ca" should be "can"

- Page 24, Figure 3 caption: More information is needed in the caption. The gray lines look like the actual averaging kernel matrix, so I'm assuming the colored lines are the row kernels? Also, what do the heights in the legend correspond to?

- Page 26, Figure 5 caption: Delete "Please"

---

## Author Comment (AC1) · 17 Apr 2018

This paper describes in detail an evaluation and error analysis of a water vapour profile retrieval algorithm (Multi-platform remote Sensing of Isotopologues for investigating the Cycle of Atmospheric water, MUSICA) that has been applied to the Infrared Atmospheric Soundings Interferometer (IASI) sensor flown onboard the EUMETSAT MetOp satellites. The algorithm is an optimal estimation (OE) algorithm after Rodgers (1990). The paper is well organized, thorough, precise and well written. As such, I recommend publication, but only after my comments/suggestions are addressed below:

We would like to thank the referee for their detailed review and valuable remarks and suggestions, which we address in the following.

**General Comments**

1. The Authors have neglected to reference any of the previous work regarding satellite sounder validation as detailed in numerous publications, especially those pertaining to the NASA Atmospheric Infrared Sounder (AIRS). In fact, I don't even recall that the AIRS sounder was even given a mention in this paper. As the Authors must know, acknowledgement of previous related work is a very important aspect of science publications. I can provide example publications, but I'm sure the Authors are already aware of them. It is important that this previous work be acknowledged in the revision.

In the introduction we give very few examples of satellite sensors that have been used for the remote sensing of atmospheric water vapour. Our idea was to give examples for sensors that use different wavelength regions (we give the examples of GOME, MODIS, TES, IASI, AMSU). We do not mention AIRS and would like to apologise for this. In the revised version we will expand the review and mention more sensors (discuss the importance of AIRS) and also discuss previous validation work.

2. On Page 7 the Authors indicate that they are using Level 2 IASI products as components in their a priori state. Unless the Authors are using independent channels from those used in the EUMETSAT L2 product, then this technically speaking would not be an a priori, which by definition is a "virtual measurement" not used prior to the retrieval. Also, this begs the question: Why not simply use the L2 $H_2O$ product? The Authors should acknowledge and clarify these considerations, and it is particularly important that they explain the latter question — else the paper's novelty and/or utility comes into question.

We use the EUMETSAT L2 atmospheric temperature as the a priori temperature of our atmospheric temperature retrievals. We constrain our atmospheric temperature retrieval by a matrix that mimics the inverse of the EUMETSAT L2 atmospheric temperature uncertainty covariance matrix (see Sect. 2.2), i.e. for our atmospheric temperature solution state we allow for variations with respect to the EUMETSAT L2 atmospheric temperatures that are in accordance to the EUMETSAT L2 atmospheric temperature uncertainties. Our atmospheric temperature product is not independent from the EUMETSAT L2 atmospheric temperatures, instead it is an update of the EUMETSAT data obtained by fitting the spectral region used in our retrieval.
The skin temperature is freely fitted, i.e. there is no constraint and the EUMETSAT L2 skin temperature is used as the first guess for the iterative retrieval method.
Our $H_2O$ (and HDO) a priori is a unique a priori for all the retrievals (no temporal and spatial variation, it is the same for all seasons and latitudes). This unique a priori is a mean profile from model calculations. In Sect 2.2 of the revised version we will give more details on the a priori information used by our retrieval.
The a priori data of the other species (N2O, CH4, HNO3, CO2) are also from model calculations and unique for all the retrievals (no temporal and spatial variation it is the same for all seasons and latitudes).

**Specific Comments**

• Page 2, Line 26: "MUSICA" — acronym should also be defined here.
Ok!

• Section 2.1: Nicely written introduction to the OE methodology — this is greatly appreciated.
Thanks!

• Page 3, Lines 3–4: This statement needs to be qualified with adequate references.
As the authors probably know, the NASA AIRS sounding algorithm (which is the pathfinder high spectral resolution infrared sounder) does not use the Rogers version of OE.
Ok, we will give references of satellite atmospheric remote sensing products generated by the optimal estimation method according to Rodgers (2000).

• Page 3, Equation (2): recommend that this be expressed as a formal equation — the "cost" term can be explicitly featured on the left side (a standard variable is J, but the authors are free to use whichever they choose).
Ok!

• Page 4, Lines 19–20: Acronym should be defined above at first occurrence.
Yes, right, we will define it in the introduction and not here.

• Page 4, Line 24: "HDO" — this chemical formula is not something I've encountered — although I (and astute readers) may deduce that it has something to do with the isotope formulas, the Authors should nevertheless define it and describe why it's relevant and/or important to this paper.
In the revised manuscript we will better explain that the MUSICA IASI processor fits $H_2O$ and HDO (actually the HDO/$H_2O$ ratio). HDO and $H_2O$ vary not fully in parallel and when fitting high resolution spectra in the thermal infrared it is important to consider the different isotopologues. Furthermore, by performing the optimal estimation of $H_2O$ and HDO/$H_2O$ we can generate a {$H_2O$,HDO/$H_2O$} pair product that is very useful for investigating moisture transport processes (e.g. David Noone, Journal of Climate 2012, doi:10.1175/JCLI-D-11-00582.1).

• Page 5, Lines 2–3: By "single a priori" do you mean a single a priori profile globally? If so, where do you obtain this?
Yes we use a unique a priori for all the retrievals (no temporal and spatial variation, it is the same for all seasons and latitudes). This unique a priori is a mean profile from model calculations.

• Page 5, Line 5: "there is no constraint on the surface temperature" — What is meant by this? What is the a priori for surface temperature?
Surface skin temperature is freely fitted and we use the EUMETSAT L2 skin temperature as the first guess when starting the iterations.

• Page 5, Lines 14–15: Although not critical to the current paper, the Authors may wish to consider more recent models in future work, such as Watts et al. (1996), Masuda (2006), or the JCSDA CRTM model (Nalli et al., 2008), and perhaps acknowledge this in the text. The Masuda et al. (1988) model is known to have significant biases at larger scan angles. Also, what is being used as the surface wind speed?
We assume a wind speed of 5 m/s. A wind speed of 0 m/s or 15 m/s would change the emissivities in the fitted 1190-1400 cm-1 wavenumber region by up to 0.5% (the effect is most severe at the maximum satellite zenith angle, which is a bit smaller than 60° in the case of IASI). However, this 0.5% emissivity error would be almost uniform over the whole fitted wavenumber region and thus has an extremely weak effect on the retrieval products. We fit a 200cm-1 broad window and uniform emissivity errors will be largely corrected by the surface skin temperature fit. If the emissivity error was varying with frequency then it would have a significant effect on the retrieval products.
Many thanks for this comment. A comparison to other models can allow an estimation of the uncertainty in the Masuda (1988) data.

• Page 6, Lines 19–20: The meaning of this sentence is not clear — I'm not sure how different time periods at the different locations means that the the dataset is not uniform.
Sect. 3.2-3.5 and Table 1 inform about the differences for the different time periods. The retrievals at Manus Island and the Lindenberg retrievals for 2008 use IASI spectra measured after October 2007. Then IASI L2 products are available and we use the IASI L2 cloud product to identify clouds and the IASI L2 atmospheric temperature product as the a priori temperature in our atmospheric temperature retrieval. This is different for the Lindenberg and Sodankylä retrievals made with spectra measured in summer 2007. For this time period no IASI L2 products are available. We use the Vaisala radiosonde: (1) together with the cloud detection algorithm of Zhang et al. (2010) for identifying clouds and (2) for creating an a priori atmospheric temperature. Because a thermal nadir retrieval product depends on the assumed a priori

temperature and is also affected by clouds (see error estimations as summarized in Figs. 7 and 9, respectively), the usage of systematically different a priori temperatures and different cloud detection algorithms can cause a significant bias between the two time periods.

• Page 6, Section 3.1: What is the source for the Vaisala sensor information? This should also be included in the references.
We are not sure if we understand this comment, because we think that we provide the relevant information in the manuscript. At the end of Sect. 3.1 we write: "We work with Vaisala RS92 data that have been processed by the GRUAN lead centre (http://www.gruan.org). The GRUAN data processing assures that the obtained humidity, pressure and temperature profiles are well-calibrated and highly accurate (Dirksen et al., 2014; Sommer et al., 2016)".

• Page 7, Line 5: I was not aware that there was a GRUAN site at Manus Island. When did this come online and is it still in operation?
Manus Island has GRUAN processed data for 2012 and 2013, but is currently inactive (see https://www.gruan.org/network/sites/)

• Page 7, Lines 8–9: See General Comment #2 above.
Yes, we use the EUMETSAT L2 atmospheric temperature as the a priori for our atmospheric temperature retrieval. However, our H2O (and HDO) a priori is a mean profile from model calculations and it is unique for all the retrievals (no temporal and spatial variation it is the same for all seasons and latitudes).

• Page 7, Lines 13–15: Why is this detail not given for the Manus site?
Yes, we will also give respective information for the Manus collocation (there the collocation has been performed by EUMETSAT).

• Page 7, Lines 24–25: I'm not sure I fully understand — if the radiosondes are being used as truth, then how can they be used as the a priori?
For Lindenberg 2007 and Sodankylä 2007 we use the radiosonde atmospheric temperature as the a priori for our atmospheric temperature retrieval. We do not use a H2O (or HDO) a priori from a measurement. Instead the used H2O (and HDO) a priori is from model calculations and it is the same for all dates and locations (it is the yellow star in the correlation plot of Fig. 11).

• Page 8, Line 5: "this altitudes" should be "these altitudes"
Ok!

• Page 12, Line 25: "about every 10 m" — I'm not sure this is correct. My understanding is that the balloon ascent is about 5 m/s, and the Vaisala processed radiosonde reports every second.
Ok, we will double check.

• Page 12, Line 27: Insert "statistically" between "performed" and "in two steps", thus "is performed statistically in two steps"
We are not sure if "statistically" really captures the method. Isn't it more that we have to fulfill two conditions: (1) partial columns should keep the same and (2) the profile should not oscillate too much?

• Page 13, Lines 3–4: Replace "In order to get the in situ profile data that are comparable to the remote sensing data we have to smooth the" simply with "The" and insert "may be smoothed" after $x_{GRUAN}$ and "according"
Ok!

• Page 14, Line 17: The logarithmic dependence of these formulations is not immediately apparent in the second lines of these equations, which simply denote relative values. I recommend inserting the intermediary mathematical step.
Ok. A similar remark has been made by referee #1. We will add the following explanation: "Because

$$\Delta \ln(x) = \frac{\Delta x}{x}$$ we interpret the logarithmic scale difference between IASI and GRUAN as the relative difference (and use GRUAN data in the denominator)."

• Page 15, Line 4: Recommend using percents here so that it is clear that one is talking about relative values.
Ok, we will use "%" for all these values.

• Page 17, Line 7: "ca" should be "can"
Ok!

• Page 24, Figure 3 caption: More information is needed in the caption. The gray lines look like the actual averaging kernel matrix, so I'm assuming the colored lines are the row kernels? Also, what do the heights in the legend correspond to?
Yes we agree. All lines show row kernels. The coloured lines are the row kernels for the altitudes as given in the legend and the gray lines depict the rest of the row kernels. We think that highlighting selected altitudes is helpful for the readability of the Figure.

• Page 26, Figure 5 caption: Delete "Please"
Ok!

---

## Author Comment (AC2) · 14 May 2018

Dear referee,

many thanks for your very useful comments, recommendations and technical corrections. Please find our reply in the attached pdf-file.

Best regards!

Please also note the supplement to this comment:

[Figure]

https://www.atmos-meas-tech-discuss.net/amt-2017-374/amt-2017-374-AC2-supplement.pdf

---

## Author Response (AR1)

We revised the manuscript in line with the recommendations from the referees. In the following we will list the most important modification with respect to the AMTD version of the manuscript. A pdf manuscript compiled with LatexDiff is attached to this "Reply to the Editor".

We have modified Section 5. The intention is to better combine the part of the manuscript that shows the error estimation with the part that presents the comparison. We now much better discuss to what extent the agreement between MUSICA IASI and GRUAN is in line with the expected errors. We provide bias, scatter and $R^2$ values for comparison of individual ensembles and for all 100 radiosondes. We empirically prove that EUMETSAT L2 temperature is indeed the leading error source and discuss possible reasons for observed biases.

We have adjusted the notation in the mathematical formulae to the recommendation of TUNER.

We very briefly discussed the limits of a comparison study using only 100 independent observations.

We better describe the collocation criteria and briefly mention the inconsistencies that can occur when using EUMETSAT L2 data belonging to different PPF software versions.

We add a reference to a publication that presents AIRS water vapour profiles and to another publication that discusses the evaluation of AIRS water vapour profiles by means of radiosonde data.

**Evaluation of MUSICA /IASI tropospheric water vapour profiles by theoretical error assessments and comparisons to GRUAN Vaisala RS92 measurements**

Christian Borger[1,a], Matthias Schneider[1], Benjamin Ertl[1,2], Frank Hase[1], Omaira E. García[3], Michael Sommer[4], Michael Höpfner[1], Stephen A. Tjemkes[5], and Xavier Calbet[6]

[1]Institute of Meteorology and Climate Research (IMK-ASF), Karlsruhe Institute of Technology, Karlsruhe, Germany
[2]Steinbuch Centre for Computing (SCC), Karlsruhe Institute of Technology, Karlsruhe, Germany
[3]Izaña Atmospheric Research Center, Agencia Estatal de Meteorología (AEMET), Santa Cruz de Tenerife, Spain
[4]Deutscher Wetterdienst, Meteorologisches Observatorium Lindenberg, Richard-Aßmann-Observatorium, Am Observatorium 12, 15848, Lindenberg/Tauche, Germany
[5]EUMETSAT, Eumetsat Allee 1, 64295 Darmstadt, Germany
[6]AEMET, C/Leonardo Prieto Castro 8, Ciudad Universitaria, 28071 Madrid, Spain
[a]now at: Satellite Remote Sensing Group, Max Planck Institute for Chemistry, Mainz, Germany

[revised manuscript text omitted]

$$J = [\boldsymbol{y} - \boldsymbol{F}(\boldsymbol{x}, \boldsymbol{b})]^T \mathbf{S}_\epsilon^{-1} \underbrace{[\boldsymbol{y} - \boldsymbol{F}(\boldsymbol{x}, \boldsymbol{p})] \mathbf{S}_{\mathbf{y,noise}}^{-1} [\boldsymbol{y} - \boldsymbol{F}(\boldsymbol{x}, \boldsymbol{b})]} + [\boldsymbol{x} - \boldsymbol{x}_a]^T \mathbf{S}_{\mathbf{a}}^{-1} [\boldsymbol{x} - \boldsymbol{x}_a]. \tag{2}$$

30  Here, the first term is a measure of the difference between the measured spectrum (represented by $\boldsymbol{y}$) and the spectrum simulated for a given atmospheric state (represented by $\boldsymbol{x}$), while taking into account the actual measurement noise  ($\mathbf{S}_{\mathbf{y,noise}}$

is the measurement noise covariance matrix). The second term of the cost function (Eq. 2) constrains the atmospheric solution state ($x$) towards an a priori most likely state ($x_a$), whereby the kind and strength of the constraint are defined by the a priori covariance matrix $\mathbf{S_a}$. The constrained solution is reached at the minimum of the cost function (Eq. 2). Due to the nonlinear behavior of $F(x, b)$, the minimisation is generally achieved iteratively. For the $(i+1)$th iteration it is:

$$x_{i+1} = x_a + \mathbf{G_i}[y - F(x_i, b) + \mathbf{K_i}(x_i - x_a)]. \tag{3}$$

$\mathbf{K}$ is the Jacobian matrix (derivatives that capture how the measurement vector will change for changes in the atmospheric state $x$). $\mathbf{G}$ is the gain matrix (derivatives that capture how the retrieved state vector will change for changes in the measurement vector $y$). $\mathbf{G}$ can be calculated from $\mathbf{K}$, $\mathbf{S_{y,noise}}$ and $\mathbf{S_a}$ as:

$$\mathbf{G} = (\mathbf{K}^T \mathbf{S_{y,noise}}^{-1} \mathbf{K} + \mathbf{S_a}^{-1})^{-1} \mathbf{K}^T \mathbf{S_{y,noise}}^{-1}. \tag{4}$$

The averaging kernel is an important component of a remote sensing retrieval and it is calculated as:

$$\mathbf{A} = \mathbf{GK}. \tag{5}$$

The averaging kernel $\mathbf{A}$ reveals how a small change of the real atmospheric state vector $x$ affects the retrieved atmospheric state vector $\hat{x}$:

$$\hat{x} - x_a = \mathbf{A}(x - x_a). \tag{6}$$

The propagation of errors due to parameter uncertainties $\Delta b$ can be estimated analytically with the help of the parameter Jacobian matrix $\mathbf{K_b}$ (derivatives that capture how the measurement vector will change for changes in the parameter $b$). According to Eq. (3), using the parameter $b + \Delta b$ (instead of the correct parameter $b$) for the forward model calculations will result in an error in the atmospheric state vector of:

$$\Delta \hat{x} = -\mathbf{G} \mathbf{K_b} \Delta b. \tag{7}$$

The respective error covariance matrix $\mathbf{S_{\hat{x},b}}$ is:

$$\mathbf{S_{\hat{x},b}} = \mathbf{G} \mathbf{K_b} \mathbf{S_b} \mathbf{K_b}^T \mathbf{G}^T, \tag{8}$$

where $\mathbf{S_b}$ is the covariance matrix of the uncertainties $\Delta b$.

Noise on the measured radiances also affects the retrievals. The error covariance matrix for noise can be analytically calculated as:

$$\mathbf{S_{\hat{x},noise}} = \mathbf{G} \mathbf{S_{y,noise}} \mathbf{G}^T, \tag{9}$$

where $\mathbf{S_{y,noise}}$ is the covariance matrix for noise on the measured radiances $y$.

**2.2 The MUSICA retrieval setup**

The MUSICA MetOp/IASI retrieval is based on a nadir version of the retrieval code PROFFIT (PROFile FIT; Hase et al., 2004) (PROFile FIT Hase et al., 2004) and on the corresponding radiative transfer model PRFFWD (PRoFit ForWarD model; Hase et al., 2004)(P The nadir code has been developed in support of the project MUSICA (MUlti-platform remote Sensing of Isotopologues for investigating the Cycle of Atmospheric water, http://www.imk-asf.kit.edu/english/musica.php). The PRFFWD-nadir code has been recently updated by including water continuum calculations according to the model "MT_CKD" v2.5.2 (Mlawer et al., 2012; Delamer

For the MUSICA MetOp/IASI retrieval calculations a single broad spectral window ranging from $1190\,\mathrm{cm}^{-1}$ to $1400\,\mathrm{cm}^{-1}$ is used. The spectral signatures of $H_2^{16}O$, $H_2^{18}O$ and $H_2^{17}O$ are fitted together as a single species and HDO (from now on called $H_2O$) and $^1H^2H^{16}O$ (from now on called HDO) as a separate species. Furthermore, the retrieval's spectral window contains spectroscopic features of $CH_4$ and $N_2O$ as well as weak spectroscopic features of $HNO_3$ and very weak spectroscopic features of $CO_2$ and $O_3$. All these trace gases (except $O_3$) are simultaneously fitted during the retrieval process whereby the spectroscopic parameters are taken from the HITRAN 2016 database (Gordon et al., 2017) with small modifications for HDO parameters (similar to Schneider et al., 2016, the line intensity parameters of HDO have been increased by 10%).

The water isotopologuesare retrieved For the water isotopologues, $CH_4$, $N_2O$ and $HNO_3$ profile retrievals are performed on a logarithmic scale. For $CO_2$ and $HNO_3$ the a priori profiles are scaledand for $CH_4$ and $N_2O$ an ad hoc regularisation is applied allowing profile retrievals on a logarithmic scale (García et al., 2017). 
[revised manuscript text omitted]
 $1185\,cm^{-1}$ and $1240\,cm^{-1}$ and $0\%$ uncertainty at the grid points $1295\,cm^{-1}$, $1350\,cm^{-1}$ and $1405\,cm^{-1}$ (this means that between $1240\,cm^{-1}$ and $1295\,cm^{-1}$ the uncertainty is linearly changing from $-1\%$ to $0\%$). The second pattern means a $-1\%$ uncertainty at the spectral grid points $1405\,cm^{-1}$ and $1350\,cm^{-1}$ and $0\%$ for the rest (with a linear change $-1\%$ to $0\%$ between $1350\,cm^{-1}$ and $1295\,cm^{-1}$). The top row of panels in Fig. B shows that surface emissivity uncertainties are mainly important for the wavenumber region below $1300\,cm{-}1$ (the first uncertainty pattern). An emissivity uncertainty of ~~+2% have a strong impact near the ground (errors of approx. 20-40% ). For Manus Island, the error is always negative in the lower troposphere and similar for all observations, whereas for Lindenberg and Sodankylä a +2% uncertainty in the surface emissivity causes sometimes positive and sometimes negative errors in the lower troposphere, i.e. the sensitivity with respect to this uncertainty source is strongly varying. Surface skin temperature is fitted during the retrieval process, i. e. uncertainties in the first guess surface skin temperatures are partly corrected for during the retrieval process. The remaining error due to an +2 K surface skin temperature uncertainty is largest in the lower troposphere, where it can reach up to 20% . It. Above 4 km altitude the surface emissivity and surface skin temperature uncertainties are smaller than 5~~
[revised manuscript text omitted]

$$_{\mathrm{GRUAN}}\hat{g} = (\mathbf{A_{11}} + \mathbf{A_{12}})(_{\mathrm{GRUAN}}g - x_a) + x_a. \tag{13}$$

Here $\mathbf{A_{11}}$ is the $H_2O$ block of the averaging kernel matrix and $\mathbf{A_{12}}$ the block that describes the response of the retrieved $H_2O$ on atmospheric HDO (see Sect. 2.3) and the vector $x_a$ is the a priori state vector. An example illustrating the effects of the regridding and the smoothing is given in Fig. 10.

We would like to note that by using Eq. (13) we assume that $H_2O$ and HDO variations are fully correlated. However, $H_2O$ and HDO do not vary fully in parallel, i.e. calculating $\hat{x}_{\mathrm{GRUAN}}$  according to Eq. (13) implies an uncertainty that can be

estimated by the uncertainty covariance matrix $\mathbf{S}_{e,\mathrm{GRUAN}}$ $\mathbf{S}_{\hat{g}}$ according to (see also Sect. 4.3 of Barthlott et al., 2017):

$$\mathbf{S}_{e,\mathrm{GRUAN}}\mathbf{S}_{\hat{g}} = \mathbf{A_{12}}\mathbf{S}_{a,\delta\mathrm{D}}\mathbf{A_{12}}^T. \tag{14}$$

Here $\mathbf{S}_{a,\delta\mathrm{D}}$ describes the actual atmospheric $\delta$D covariances. Because $\mathbf{A_{12}}$ and $\mathbf{S}_{a,\delta\mathrm{D}}$ have small entries only, this uncertainty is below 1% and can be neglected for our comparison.

**5.2  Metric for quantifying data agreement**

 For a better statistical quantification of the deviations of the remote sensing data from the GRUAN reference data, we introduce a skill score DL describing the difference of the logarithmic values of the respective water vapour concentrations. Because $\Delta\ln(x) \approx \frac{\Delta x}{x}$, we interpret the logarithmic scale difference between IASI and GRUAN as the relative difference (and use the GRUAN data in the denominator). DL then becomes:

$$
\begin{aligned}
\mathrm{DL} &= \ln\left([\mathrm{H_2O}]_{\mathrm{retrieval}}\right) - \ln\left([\mathrm{H_2O}]_{\mathrm{GRUAN}}\right) \\
&\approx \frac{[\mathrm{H_2O}]_{\mathrm{retrieval}} - [\mathrm{H_2O}]_{\mathrm{GRUAN}}}{[\mathrm{H_2O}]_{\mathrm{GRUAN}}},
\end{aligned}
\tag{15}
$$

where $[\mathrm{H_2O}]_{\mathrm{GRUAN}}$ is the regridded and smoothed  radiosonde $\mathrm{H_2O}$ data (i.e. $\hat{g}$ from Eq. 13) and $[\mathrm{H_2O}]_{\mathrm{retrieval}}$ is the retrieved IASI $\mathrm{H_2O}$ data. The so defined skill score DL is a good measure for the relative difference between the GRUAN and IASI data.

As a good measure for the mean relative difference between GRUAN and IASI we can use the mean difference of logarithmic values (MDL):

$$
\begin{aligned}
\mathrm{MDL} = \frac{1}{N}\sum_{i=1}^{N}\mathrm{DL}_i &= \frac{1}{N}\sum_{i=1}^{N}\left[\ln\left([\mathrm{H_2O}]_{\mathrm{retrieval}}\right) - \ln\left([\mathrm{H_2O}]_{\mathrm{GRUAN}}\right)\right]_i \\
&\approx \frac{1}{N}\sum_{i=1}^{N}\left(\frac{[\mathrm{H_2O}]_{\mathrm{retrieval}} - [\mathrm{H_2O}]_{\mathrm{GRUAN}}}{[\mathrm{H_2O}]_{\mathrm{GRUAN}}}\right)_i.
\end{aligned}
\tag{16}
$$

Similarly, we can use the standard deviation of the logarithmic differences as a measure for the relative scatter between GRUAN and IASI and introduce $\sigma_{\mathrm{MDL}}$ as

$$\sigma_{\mathrm{MDL}} = \sqrt{\frac{1}{N}\sum_{i=1}^{N}(\mathrm{DL}_i - \mathrm{MDL})^2}. \tag{17}$$

For illustrating the variation of the atmospheric state we introduce $\sigma_{\hat{g}}$ as

$$\sigma_{\hat{g}} = \sqrt{\frac{1}{N}\sum_{i=1}^{N}\left[\ln\left([\mathrm{H_2O}]_{\mathrm{GRUAN}}\right)_i - \overline{\ln\left([\mathrm{H_2O}]_{\mathrm{GRUAN}}\right)}\right]^2}. \tag{18}$$

We want to document to what extent the differences between GRUAN and MUSICA IASI data can be explained by the estimated errors. In Sect. 4 we estimate in detail the error in the MUSICA IASI H$_2$O profiles for three different climate zones. It should be noted that the radiosonde measurements are also affected by several uncertainties which have to be taken into account during the evaluation.

[revised manuscript text omitted]

20  ### 5.3.3

$$
\begin{aligned}
\underline{DL} &= \underline{\ln\left([H_2O]_{\mathrm{retrieval}}\right) - \ln\left([H_2O]_{\mathrm{GRUAN}}\right)} \\
&\approx \underline{\frac{[H_2O]_{\mathrm{retrieval}} - [H_2O]_{\mathrm{GRUAN}}}{[H_2O]_{\mathrm{GRUAN}}}},
\end{aligned}
$$

25  An upper tropospheric dry bias is consistently observed in the analysis of the LI08, LI07 and

As a good measure for the mean relative difference between GRUAN and IASI we can use the mean difference of logarithmic values (MDL):

$$\mathrm{MDL} = \frac{1}{N}\sum_{i=1}^{N}\mathrm{DL}_i \quad\equiv\quad \frac{1}{N}\sum_{i=1}^{N}\left[\ln\left([H_2O]_{\mathrm{retrieval}}\right) - \ln\left([H_2O]_{\mathrm{GRUAN}}\right)\right]_i$$

$$\approx \quad \frac{1}{N}\sum_{i=1}^{N}\left(\frac{[H_2O]_{\mathrm{retrieval}} - [H_2O]_{\mathrm{GRUAN}}}{[H_2O]_{\mathrm{GRUAN}}}\right)_i .$$

5    Similarly, we can use the standard deviation of the logarithmic differences as a measure for the relative scatter between GRUAN and IASI and introduce $\sigma_{\mathrm{MDL}}$ as

$$\sigma_{\mathrm{MDL}} = \sqrt{\frac{1}{N}\sum_{i=1}^{N}\left(\mathrm{DL}_i - \mathrm{MDL}\right)^2} .$$

For illustrating the variation of the atmospheric state we introduce $\sigma_{\mathrm{GRUAN}}$ as

$$\sigma_{\mathrm{GRUAN}} = \sqrt{\frac{1}{N}\sum_{i=1}^{N}\left[\ln\left([H_2O]_{\mathrm{GRUAN}}\right)_i - \overline{\ln\left([H_2O]_{\mathrm{GRUAN}}\right)}\right]^2} .$$

10    SK07 ensembles, but not seen in the analysis of the MI ensemble. A systematic uncertainty source that affects upper tropospheric $H_2O$ at Lindenberg and Sodankylä but not at Manus Island is the shape of the water vapour lines (see discussion in the context of Fig. B). So deficits in simulating the line shapes might explain this upper tropospheric dry bias. In the near surface atmosphere we observe a wet bias at the two continental sites Lindenberg and Sodankylä, but only for the ensembles that are limited to the summer season (LI07 and SK07). Our error estimation study suggests that small uncertainties in the emissivity can cause

15    large errors at these continental sites. So an uncertainty in the used IREMIS emissivity is a candidate for explaining the surface near wet bias; however, the $H_2O$ retrieval response for a $-1\,\%$ uncertainty in the emissivity differs betweeen observations and can be positive or negative (see Fig. 9). This means that emissivity uncertainties can only explain the bias if the sign of the emissivity uncertainty is correlated with the atmospheric state (e.g. the uncertainty in the used monthly IREMIS surface emissivity is typically positive for dry atmospheric conditions and typically negative for humid atmospheric conditions) or

20    surface conditions (e.g. the uncertainty in the IREMIS data is typically positive/negative for a surface with high/low emissivity or high/low skin temperatures).
    Figure ??

**5.4    Global overview on data agreement**

Figure 13 depicts the vertical profiles of the aforementioned skill scores calculated data agreement skill score parameters for all

25    coinciding observations without separating the different sites and time periods. The MDL value is within ±0.12 for all altitudes , indicating that the retrieval is in very good agreement to 
[revised manuscript text omitted]

[Figure]

**Figure 2.** Vertical H$_2$O profiles as measured by the 100 different GRUAN processed Vaisala RS92 radiosondes: from Manus Island, Linden-berg and Sodankylä used for our study. Black lines indicate radiosonde data ensembles that cover all seasons (Manus Island and Lindenberg 2008) and red lines indicate ensembles that cover the summer season only.

[Figure]

**Figure 3.** Example row kernels ($\mathbf{A_{11}} + \mathbf{A_{12}}$, see Eq. 11) for the three reference sites. Manus Island: 2013-11-28 11:37:24 UT, satellite zenith angle 12.4°,  precipitable water vapour 46.4 mm; Lindenberg: 2008-10-08 20:00:38 UT, 16.3°, 14.1 mm; Sodankylä: 2007-08-24 08:31:25 UT, 27.8°, 13.7 mm. Numbers in the upper right corners of every panel indicate the respective degrees of freedom of signal (DOFS). Row kernels of selected altitudes are highlighted by thick coloured lines. The thick black dashed line represents the sum along the row of the averaging kernel matrix.

[Figure]

**Figure 4.** Variation of the DOFS values ( degrees of freedom for signal) for the four different ensembles.

[Figure]

**Figure 5.**  Same as  Fig. 3   but for   a typical winter and summer observation above Lindenberg.  Summer observation : 2008-08-01 08:39:01 UT,  satellite zenith angle 43.7°, precipitable water vapour 28.8 mm; Winter observation: 2008-02-15 09:54:30 UT, 41.2°, 3.2 mm.

[Figure]

**Figure 6.** Spectral responses of uncertainty sources for a typical situation at Manus Island (same situation as for the kernel in Fig. 3). The left panel shows examples for the influence of uncertainties in temperatures (surface skin $\Delta T = +2\,\mathrm{K}$, lower tropospheric $\Delta T = +2\,\mathrm{K}$ and upper tropospheric $\Delta T = +1\,\mathrm{K}$). The right panel illustrates examples for the influence of clouds (dust layer (4-6 km) and cirrus cloud (13-14 km and 50% cloud fraction)) on the spectrum. Note the different y-axis scales, i.e. the positive response for positive temperature uncertainties and the negative response for unrecognized clouds.

[Figure]

**Figure 7.** H$_2$O error profiles  derived from the random uncertainty sources: instrument noise, emissivity and atmospheric temperatures. Shown are the square root  values of the diagonal of the  matrices $\mathbf{S}_{\hat{\mathbf{x}},\mathrm{noise}}$ (for instrument noise) and $\mathbf{S}_{\hat{\mathbf{x}},b}$ (for emissivity and atmospheric temperatures) according to Eqs. (9) and (8), respectively.  The data are  depicted for  for all members of the MI, LI08, and SK07 ensembles.

[Figure]

Same as 1.  and 1 K in the  other layers), and spectroscopy (line strength, +5%, and pressure broadening, +5%) and the water vapour continuum (assuming a 10% underestimation of the "MT_CKD" model). Shown are the errors in the atmospheric state vector $\Delta\hat{x}$ according to Eqs. (7). The data are depicted for for all members of the MI, LI08, and SK07 ensembles.

**Figure 8.** H₂O error  profiles $x_e$ **37** derived from the systematic uncertainty sources: emissivity (-1% in two different wavenumber regions), atmospheric  temperatures ( 2 K between surface and 2 km

[Figure]

**Figure 9.** Same as Fig. B, but for errors due to unrecognized clouds: cirrus  (50% fractional coverage location of cloud layers see legend) cumulus  (10% fractional coverage with cloud top altitudes as given in the legends) and mineral dust (homogeneous dust clouds  layers as  give in the legends and with the composition according to OPAC "Desert").

[Figure]

**Figure 10.** Example for the regridding and smoothing of the GRUAN data required before validating the MUSICA /IASI retrieval of
$H_2O$ profiles. Black line: raw GRUAN data; Red line: regridded GRUAN data ($x_{GRUAN}g$); Green line: regridded and smoothed radiosonde
data ($\hat{x}_{GRUAN}\hat{g}$, according to Eq. 13).

[Figure]

Correlation between GRUAN (along y-axes) and MUSICA MetOp/IASI data (along x-axes) at the 3 different reference sites for 3 different atmospheric levels (lower, middle and upper troposphere). The yellow star represents the a priori assumption for the respective retrieval level (the retrieval uses globally the same a priori). The respective retrieval level altitudes are given in the individual scatter plots. Red and black colour distinguish the remote sensing data ensembles that use different input data (MI and LI08, on the one hand, and LI07 and SK07, on the other hand; see Sect. 3.1 and Tab. 1). The blue dotted line represents the 1-to-1 diagonal and the blue error bars indicate the typical GRAUN and IASI errors.

**Figure 11.** Vertical profiles of retrieval skill scores calculated according to Eqs. (16) and using data from all ensembles ((19) for the MI , LI07, LI08 and SK07)LI08 ensembles. MDL (red The black line ) is and error bars represent the mean difference and the $1\sigma$ scatter between IASI and smoothed GRUAN data ; $\pm\sigma_{MDL}$ (blue lines MDL and blue $\pm\sigma_{MDL}$). The red shaded area ) around MDL is the $1\sigma$ scatter between expected due to MUSICA IASI and smoothed GRUAN data; $\pm\sigma_{GRUAN}$ errors (black lines $\pm\Delta_{MDL}$)is . The grey shaded area represents the area beyond the $1\sigma$ variability of smoothed GRUAN data (area beyond $\pm\sigma_{\hat{g}}$).

[Figure]

**Figure 12.** Same as Fig.11, but for MUSICA IASI retrievals that use the GRUAN temperature profiles as the a priori atmospheric temperatures and all four ensembles (MI, LI08, LI07 and SO).

[Figure]

**Figure 13.** Same as Fig.12, but considering all four ensembles as a single data set.

[Figure]

**Figure 14.** Correlation between GRUAN (along x-axes) and MUSICA IASI data (along y-axes) for the 6 different atmospheric altitudes that are highlighted in Figs. 3 and 5. All the presented data are for MUSICA IASI retrievals that use the GRUAN temperature profiles as the a priori atmospheric temperatures. The retrieval altitudes are given in the individual scatter plots, together with correlation coefficient ($R^2$), bias (b) and scatter (s). Data belonging to the different ensembles can be identified by the symbols and colours as described in the legend (bottom right). The yellow stars represent the a priori value (the retrieval uses globally the same $H_2O$ a priori) and the blue error bars indicate the typical GRAUN and IASI errors. The dotted line represents is the 1-to-1 diagonal.

[Figure]

**Figure 15.** Profiles of correlation coefficients ($R^2$) for comparison between GRUAN and MUSICA IASI (for retrievals that use the GRUAN temperature profiles as the a priori atmospheric temperatures). Different line colours and symbols for the different ensembles and thick black solid line for considering all four ensembles as a single data set (the $R^2$ values for the latter are also written in the panels of Fig. 14).

[Figure]

**Figure A1.** Profiles of the WVMR errors of the GRUAN radiosondes: The top panels represent the correlated errors and the bottom panels the uncorrelated errors. The colours distinguish the different ensembles of the retrieval setup: black for MI and LI08, red for LI07 and SK07.

[Figure]

**Figure A2.** Same as top panels of Fig. A1, but for the correlated errors in the regridded and smoothed GRUAN radiosonde data.

[Figure]

**Figure A3.** Profiles of the correlated temperature errors of the GRUAN radiosondes. Above the radiosondes top height we use a zonally and monthly mean temperature climatology and assume an uncertainty of 5 K.

**Table 1.** Overview  on the different ensembles of GRUAN reference and the available retrieval input data.

| | Manus Island | Lindenberg 2008 | Lindenberg 2007 | Sodankylä 2007 |
|---|---|---|---|---|
|  Acronym | MI | LI08 | LI07 | SK07 |
| Time period | 2011-2013 | 2008 (all months) | 2007 ( June-August) | 2007 ( June-August) |
|  Number of independent GRUAN sondes | 25 | 32 | 26 | 17 |
| Ground level | EUMETSAT IASI L2 (GTOPO30) | EUMETSAT IASI L2 (GTOPO30) | GTOPO30 | GTOPO30 |
|  Emissivity | EUMETSAT IASI L2 (Masuda et al., 1988) | EUMETSAT IASI L2 (IREMIS) | IREMIS | IREMIS |
|  Cloud identification | EUMETSAT IASI L2 + visual inspection | EUMETSAT IASI L2 | Zhang et al. (2010) | Zhang et al. (2010) |
|  A priori for atmospheric and surface skin temperature | EUMETSAT IASI L2 (PPF v5) | EUMETSAT IASI L2 (PPF v4) |  GRUAN sonde |  GRUAN sonde |

**Table 2.** List of uncertainty assumptions used for the error estimation of the MUSICA IASI water vapour product (for emissivity and atmospheric temperature we assume random and systematic uncertainties).

| Source | Type | Value |
|---|---|---|
| Instrumental noise | Random (Gaussian pdf) | Noise covariance according to Pequignot et al. (2008) |
| Surface emissivity | Random (Gaussian pdf) + Systematic | Random: 1% with spectral frequency correlation length of $100\,\text{cm}^{-1}$; Systematic: -1% for $<1295\,\text{cm}^{-1}$ and -1% for $>1295\,\text{cm}^{-1}$ |
| EUMETSAT L2 atmos. temp. | Random (Gaussian pdf) + Systematic | Random: 2 K from ground-2 km and 1 K above 2 km altitude with correlation length increasing from 2 km at ground to 10 km in the stratosphere; Systematic: 2 K for ground-2 km, and 1 K for 2-5 km, 5-10 km and 10 km-TOA |
| Water vapour continuum | Systematic | 10% of model MT_CKD v2.5.2 |
| Line intensity $H_2O$ and HDO | Systematic | 5% |
| Pressure-broadening $H_2O$ and HDO | Systematic | 5% |
| Opaque cumulus cloud | Random (unknown pdf) but systematic sign | 10% fractional cover with cloud top at 1.3, 3.0 and 4.9 km |
| Cirrus cloud | Random (unknown pdf) but systematic sign | Particle properties according to OPAC "Cirrus 3", 1 km thickness, 50% fractional cover with cloud top at 6, 8, 11 and 14 km |
| Mineral dust cloud | Random (unknown pdf) but systematic sign | Particle properties according to OPAC "Desert", homogeneous coverage for layers: ground-2 km, 2-4 km and 4-6 km |